# Mechanistic Interpretability for AI Safety
# A Review

**Leonard Bereska**       **Efstratios Gavves**
*{leonard.bereska, egavves}@uva.nl*
*University of Amsterdam*

**Reviewed on OpenReview:** *https://openreview.net/forum?id=ePUVetPKu6*

## Abstract

Understanding AI systems' inner workings is critical for ensuring value alignment and safety. This review explores mechanistic interpretability: reverse engineering the computational mechanisms and representations learned by neural networks into human-understandable algorithms and concepts to provide a granular, causal understanding. We establish foundational concepts such as features encoding knowledge within neural activations and hypotheses about their representation and computation. We survey methodologies for causally dissecting model behaviors and assess the relevance of mechanistic interpretability to AI safety. We examine benefits in understanding, control, alignment, and risks such as capability gains and dual-use concerns. We investigate challenges surrounding scalability, automation, and comprehensive interpretation. We advocate for clarifying concepts, setting standards, and scaling techniques to handle complex models and behaviors and expand to domains such as vision and reinforcement learning. Mechanistic interpretability could help prevent catastrophic outcomes as AI systems become more powerful and inscrutable. For an HTML version of the paper, visit https://leonardbereska.github.io/blog/2024/mechinterpreview/.

## 1 Introduction

As AI systems rapidly become more sophisticated and general (Bubeck et al., 2023; Bengio et al., 2023), advancing our understanding of these systems is crucial to ensure their alignment (Ji et al., 2024) with human values and avoid catastrophic outcomes (Hendrycks et al., 2023; Hendrycks & Mazeika, 2022). The field of interpretability aims to demystify the internal processes of AI models, moving beyond evaluating performance alone. This review focuses on mechanistic interpretability, an emerging approach within the broader interpretability landscape that strives to comprehensively specify the computations underlying deep neural networks. We emphasize that understanding and interpreting these complex systems is not merely an academic endeavor – *it's a societal imperative to ensure AI remains trustworthy and beneficial.*

The interpretability landscape is undergoing a paradigm shift akin to the evolution from behaviorism to cognitive neuroscience in psychology. Historically, lacking tools for introspection, psychology treated the mind as a black box, focusing solely on observable behaviors. Similarly, interpretability has predominantly relied on black-box techniques (Casper et al., 2024), analyzing models based on input-output relationships or using attribution methods that, while probing deeper, still neglect the model's internal architecture. However, just as advancements in neuroscience allowed for a deeper understanding of internal cognitive processes, the field of interpretability is now moving towards a more granular approach. This shift from surface-level analysis to a focus on the internal mechanics of deep neural networks characterizes the transition towards inner interpretability (Räuker et al., 2023).

Mechanistic interpretability, as an approach to inner interpretability, aims to completely specify a neural network's computation, potentially in a format as explicit as pseudocode (also called *reverse engineering*), striving for a granular and precise understanding of model behavior. It distinguishes itself primarily through

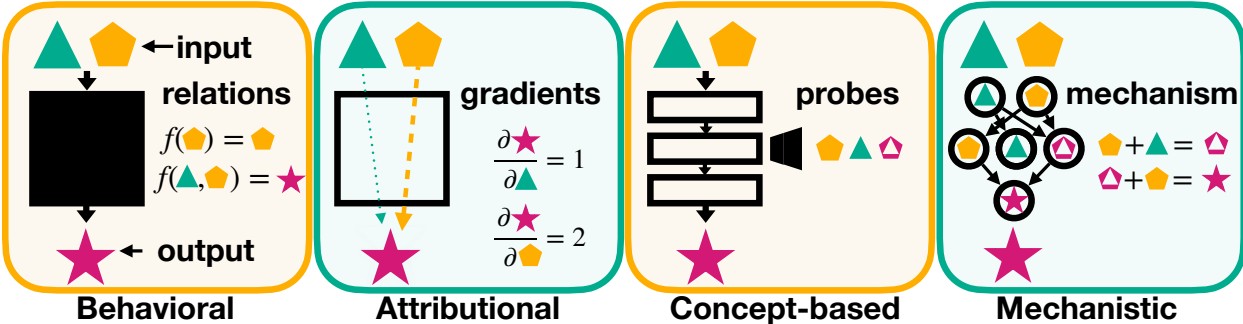

Figure 1: Interpretability paradigms offer distinct lenses for understanding neural networks: **Behavioral** analyzes input-output relations; **Attributional** quantifies individual input feature influences; **Concept-based** identifies high-level representations governing behavior; **Mechanistic** uncovers precise causal mechanisms from inputs to outputs.

its *ambition* for comprehensive reverse engineering and its strong *motivation* towards AI safety. Our review serves as the first comprehensive exploration of mechanistic interpretability research, with the most accessible introductions currently scattered in a blog or list format (Olah, 2022; Nanda, 2022d; Olah et al., 2020; Sharkey et al., 2022a; Olah et al., 2018; Nanda, 2023f; 2024). Concurrently, Ferrando et al. (2024) and Rai et al. (2024) have also contributed valuable reviews giving concise, technical introductions to mechanistic interpretability in transformer-based language models. Our work complements these efforts by synthesizing the research (addressing the "research debt" (Olah & Carter, 2017)) and providing a structured, accessible, and comprehensive introduction for AI researchers and practitioners.

The structure of this paper provides a cohesive overview of mechanistic interpretability, situating the mechanistic approach in the broader interpretability landscape (Section 2), presenting core concepts and hypotheses (Section 3), explaining methods and techniques (Section 4), presenting a taxonomy and survey of the current field (Section 5), exploring relevance to AI safety (Section 6), and addressing challenges (Section 7) and future directions (Section 8).

## 2 Interpretability Paradigms from the Outside In

We encounter a spectrum of interpretability paradigms for decoding AI systems' decision-making, ranging from external black-box techniques to internal analyses. We contrast these paradigms with mechanistic interpretability, highlighting its distinct causal bottom-up perspective within the broader interpretability landscape (see Figure 1).

**Behavioral** interpretability treats the model as a black box, analyzing input-output relations. Techniques such as minimal pair analysis (Warstadt et al., 2020), sensitivity and perturbation analysis (Casalicchio et al., 2018) examine input-output relations to assess the model's robustness and variable dependencies (Shapley, 1988; Ribeiro et al., 2016; Covert et al., 2021). Its *model-agnostic* nature is practical for complex or proprietary models but lacks insight into internal decision processes and causal depth (Jumelet, 2023).

**Attributional** interpretability aims to explain outputs by tracing predictions to individual input contributions using gradients. Raw gradients can be discontinuous or sensitive to slight perturbations. Therefore, techniques such as SmoothGrad (Smilkov et al., 2017) and Integrated Gradients (Sundararajan et al., 2017) average across gradients. Other popular techniques are layer-wise relevance propagation (Bach et al., 2015), DeepLIFT (Shrikumar et al., 2017), or GradCAM (Selvaraju et al., 2016). Attribution enhances transparency by showing input feature influence without requiring an understanding of the internal structure, enabling decision validation, compliance, and trust while serving as a bias detection tool, but also has fundamental limitations (Bilodeau et al., 2024).

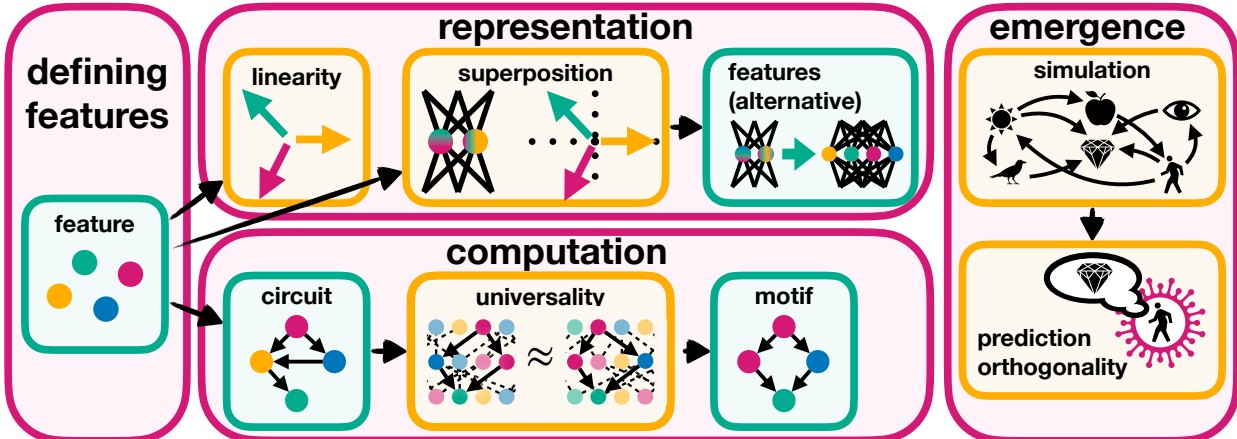

Figure 2: Overview of key concepts and hypotheses in mechanistic interpretability, organized into four subsection (pink boxes): defining features (Section 3.1), representation (Section 3.2), computation (Section 3.3), and emergence (Section 3.4). In turquoise, it highlights definitions like *features*, *circuits*, and *motifs*, and in orange, it highlights hypotheses like *linear representation*, *superposition*, *universality*, *simulation*, and *prediction orthogonality*. Arrows show relationships, *e.g.*, superposition enabling an alternative feature definition or universality connecting circuits and motifs.

**Concept-based** interpretability adopts a top-down approach to unraveling a model's decision-making processes by probing its learned representations for high-level concepts and patterns governing behavior. Techniques include training supervised auxiliary classifiers (Belinkov, 2021), employing unsupervised contrastive and structured probes (see Section 4.2) to explore latent knowledge (Burns et al., 2023), and using neural representation analysis to quantify the representational similarities between the internal representations learned by different neural networks (Kornblith et al., 2019; Bansal et al., 2021). Beyond observational analysis, concept-based interpretability can enable manipulation of these representations – also called *representation engineering* (Zou et al., 2023) – potentially enhancing safety by upregulating concepts such as honesty, harmlessness, and morality.

**Mechanistic** interpretability is a bottom-up approach that studies the fundamental components of models through granular analysis of features, neurons, layers, and connections, offering an intimate view of operational mechanics. Unlike concept-based interpretability, it aims to uncover causal relationships and precise computations transforming inputs into outputs, often identifying specific neural circuits driving behavior. This *reverse engineering* approach draws from interdisciplinary fields like physics, neuroscience, and systems biology to guide the development of transparent, value-aligned AI systems. Mechanistic interpretability is the primary focus of this review.

## 3 Core Concepts and Assumptions

This section introduces the key concepts and hypotheses of mechanistic interpretability, as summarized in Figure 2. We start by defining features as the basic units of representation (Section 3.1). We then examine the nature of these features, including the challenges posed by polysemantic neurons and the implications of the superposition and linear representation hypotheses (Section 3.2). Next, we explore computation through circuits and motifs, considering the universality hypothesis (Section 3.3). Finally, we discuss the implications for understanding emergent properties, such as internal world models and simulated agents with potentially misaligned objectives (Section 3.4).

### 3.1 Defining Features as Representational Primitives

**Features as fundamental units of representation.** The notion of a *feature* in neural networks is central yet elusive, reflecting the pre-paradigmatic state of mechanistic interpretability. We adopt the notion of *features* as the *fundamental units of neural network representations*, such that features cannot be further *disentangled* into simpler, distinct factors. These features are core components of a neural network's representation, analogous to how cells form the fundamental unit of biological organisms (Olah et al., 2020).

> **Definition 1: Feature**
>
> Features are the fundamental units of neural network representations that cannot be further decomposed into simpler independent factors.

**Concepts as natural abstractions.** The world consists of various entities that can be grouped into categories or *concepts* based on shared properties. These concepts form high-level summaries like "tree" or "velocity," allowing compact world representations by discarding many irrelevant low-level details. Neural networks can capture and represent such *natural abstractions* (Chan et al., 2023) through their learned *features*, which serve as building blocks of their internal representations, aiming to capture the *concepts* underlying the data.

**Features encoding input patterns.** In traditional machine learning, *features* are understood as characteristics or attributes derived directly from the input data stream (Bishop, 2006). This view is particularly relevant for systems focused on *perception*, where features map closely to the input data. However, in more advanced systems capable of *reasoning* with abstractions, features may emerge internally within the model as representational patterns, even when processing information unrelated to the input. In this context, features are better conceptualized as *any measurable property or characteristic of a phenomenon* (Olah, 2022), encoding abstract concepts rather than strictly reflecting input attributes.

**Features as representational atoms.** A key property of features is their irreducibility, meaning they cannot be decomposed into or expressed as a combination of simpler, independent factors. In the context of input-related features, Engels et al. (2024) define a feature as *irreducible* if it cannot be decomposed into or expressed as a combination of statistically independent patterns or factors in the original input data. Specifically, a feature is reducible if transformations reveal its underlying pattern, which can be separated into independent co-occurring patterns or is a mixture of patterns that never co-occur. We propose generalizing this notion of irreducibility to features encoding abstract concepts not directly tied to input patterns, such that features cannot be reduced to combinations or mixtures of other independent components within the model's representations.

**Features beyond human interpretability.** Features could be defined from a *human-centric perspective* as *semantically meaningful, articulable input patterns encoded in the network's activation space* (Olah, 2022). However, while cognitive systems may converge on similar *natural abstractions* (Chan et al., 2023), these need not necessarily align with human-interpretable *concepts*. Adversarial examples have been interpreted as non-interpretable features meaningful to models but not humans. Imperceptible perturbations fool networks, suggesting reliance on alien representational patterns (Ilyas et al., 2019). As models surpass human capabilities, their learned features may become increasingly abstract, encoding information in ways incongruent with human intuition (Hubinger, 2019a). Mechanistic interpretability aims to uncover the *actual* representations learned, even if diverging from human concepts. While human-interpretable concepts provide guidance, a non-human-centric perspective that defines features as independent model components, whether aligned with human concepts or not, is a more comprehensive and future-proof approach.

### 3.2 Nature of Features: From Monosemantic Neurons to Non-Linear Representations

**Neurons as Computational Units?** In the architecture of neural networks, neurons are the natural computational units, potentially representing individual features. Within a neural network representation

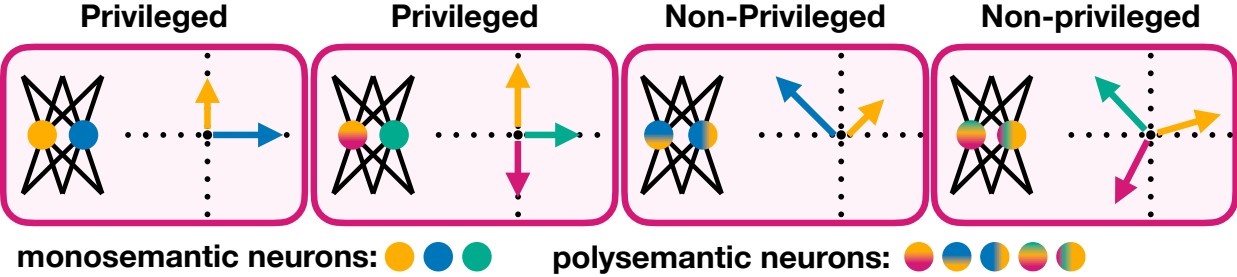

Figure 3: Contrasting privileged and non-privileged bases. In a non-privileged basis, there is no reason to expect features to be basis-aligned – calling basis dimensions neurons has no meaning. In a privileged basis, the architecture treats basis directions differently – features can but need not align with neurons (Bricken et al., 2023). **Leftmost:** Privileged basis; individual features (arrows) align with basis directions, resulting in *monosemantic* neurons (colored circles). **Middle left:** Privileged basis, where despite having more features than neurons, some neurons are monosemantic, representing individual features, while others are *polysemantic* (overlapping gradients), encoding *superposition* of multiple features. **Middle right:** Non-privileged basis where, even when the number of features equals the number of neurons, the lack of alignment between the feature directions and basis directions results in polysemantic neurons encoding combinations of features. **Rightmost:** Non-privileged, polysemantic neurons as feature directions do not align with neuron basis.

$h \in \mathbb{R}^n$, the $n$ basis directions are called neurons. For a neuron to be meaningful, the basis directions must functionally differ from other directions in the representation, forming a *privileged basis* – where the basis vectors are architecturally distinguished within the neural network layer from arbitrary directions in activation space, as shown in Figure 3. Typical non-linear activation functions privilege the basis directions formed by the neurons, making it meaningful to analyze individual neurons (Elhage et al., 2022b). Analyzing neurons can give insights into a network's functionality (Sajjad et al., 2022; Mu & Andreas, 2020; Dai et al., 2022; Ghorbani & Zou, 2020; Voita et al., 2023; Durrani et al., 2020; Goh et al., 2021; Bills et al., 2023; Huang et al., 2023).

**Monosemantic and Polysemantic Neurons.** A neuron corresponding to a single semantic concept is called *monosemantic*. The intuition behind this term comes from analyzing what inputs activate a given neuron, revealing its associated semantic meaning or concept. If neurons were the representational primitives of neural networks, all neurons would be monosemantic, implying a one-to-one relationship between neurons and features. Comprehensive interpretability would be as tractable as characterizing all neurons and their connections. However, empirically, especially for transformer models (Elhage et al., 2022b), neurons are often observed to be *polysemantic*, *i.e.*, associated with multiple, unrelated concepts (Arora et al., 2018; Mu & Andreas, 2020; Elhage et al., 2022a; Olah et al., 2020). For example, a single neuron may be activated by both images of cats and images of cars, suggesting it encodes multiple unrelated concepts. Polysemanticity contradicts the interpretation of neurons as representational primitives and, in practice, makes it challenging to understand the information processing of neural networks.

**Exploring Polysemanticity: Hypotheses and Implications.** To understand the widespread occurrence of polysemanticity in neural networks, several hypotheses have been proposed:

*i.)* One trivial scenario would be that feature directions are orthogonal but not aligned with the basis directions (neurons). There is no inherent reason to assume that features would align with neurons in a non-privileged basis, where the basis vectors are not architecturally distinguished. However, even in a privileged basis formed by the neurons, the network could represent features not in the standard basis but as linear combinations of neurons (see Figure 3, middle right).

*ii.)* An alternative hypothesis posits that *redundancy due to noise* introduced during training, such as random dropout (Srivastava et al., 2014), can lead to redundant representations and, consequently,

to polysemantic neurons (Marshall & Kirchner, 2024). This process involves distributing a single feature across several neurons rather than isolating it into individual ones, thereby encouraging polysemanticity.

*iii.)* Finally, the *superposition* hypothesis addresses the limitations in the network's representative capacity – the number of neurons versus the number of crucial concepts. This hypothesis argues that the limited number of neurons compared to the vast array of important concepts necessitates a form of compression. As a result, an $n$-dimensional representation may encode features not with the $n$ basis directions (neurons) but with the $\propto \exp(n)$ possible almost orthogonal directions (Elhage et al., 2022b), leading to polysemanticity.

---

**Hypothesis 1: Superposition**

Neural networks represent more features than they have neurons by encoding features in overlapping combinations of neurons.

---

**Superposition Hypothesis.** The *superposition* hypothesis suggests that neural networks can leverage high-dimensional spaces to represent more features than their actual neuron count by encoding features in almost orthogonal directions. Non-orthogonality means that features interfere with one another. However, the benefit of representing many more features than neurons may outweigh the interference cost, mainly when concepts are sparse and non-linear activation functions can error-correct noise (Elhage et al., 2022b).

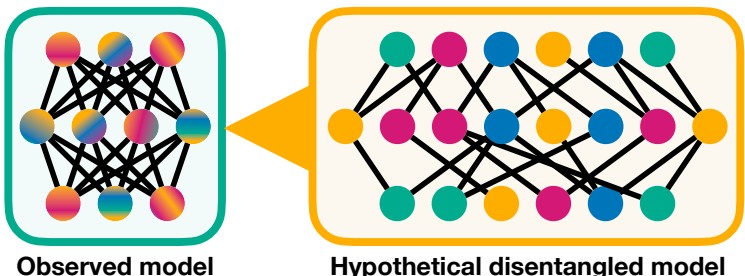

**Observed model**       **Hypothetical disentangled model**

Figure 4: Observed neural networks (left) can be viewed as compressed simulations of larger, sparser networks (right) where neurons represent distinct features. An "almost orthogonal" projection compresses the high-dimensional sparse representation, manifesting as polysemantic neurons involved with multiple features in the lower-dimensional observed model, reflecting the compressed encoding. Figure adapted from (Bricken et al., 2023).

Toy models can demonstrate under which conditions superposition occurs (Elhage et al., 2022b; Scherlis et al., 2023). Neural networks, via superposition, may effectively simulate computation with more neurons than they possess by allocating each feature to a linear combination of neurons, creating what is known as an overcomplete linear basis in the representation space. This perspective on superposition suggests that polysemantic models could be seen as compressed versions of hypothetically larger neural networks where each neuron represents a single concept (see Figure 4). Consequently, an alternative definition of features could be:

---

**Definition 2: Feature (Alternative)**

Features are elements that a network would ideally assign to individual neurons if neuron count were not a limiting factor (Bricken et al., 2023). In other words, *features* correspond to the disentangled *concepts* that a larger, sparser network with sufficient capacity would learn to represent with individual neurons.

---

## Toy Model of Superposition

A toy model (Elhage et al., 2022b) investigates the hypothesis that neural networks can represent more *features* than the number of neurons by encoding real-world *concepts* in a compressed manner. The model considers a high-dimensional vector $\boldsymbol{x}$, where each element $x_i$ corresponds to a feature capturing a real-world concept, represented as a random vector with varying importance determined by a weight $a_i$. These features are assumed to have the following properties:

   *i.)* **Concept sparsity**: Real-world concepts occur sparsely.

   *ii.)* **More concepts than neurons**: The number of potential concepts vastly exceeds the available neurons.

   *iii.)* **Varying concept importance**: Some concepts are more important than others for the task at hand.

The input vector $\boldsymbol{x}$ represents features capturing these concepts, defined by a sparsity level $S$ and an importance level $a_i$ for each feature $x_i$, reflecting the sparsity and varying importance of the underlying concepts. The model dynamics involve transforming $\boldsymbol{x}$ into a hidden representation $\boldsymbol{h}$ of lower dimension, and then reconstructing it as $\boldsymbol{x}'$:

$$\boldsymbol{h} = \boldsymbol{W}\boldsymbol{x}, \quad \boldsymbol{x}' = \mathrm{ReLU}(\boldsymbol{W}^\top \boldsymbol{h} + \boldsymbol{b}).$$

The network's performance is evaluated using a loss function $\mathcal{L}$ weighted by the feature importances $a_i$, reflecting the importance of the underlying concepts:

$$\mathcal{L} = \sum_{\boldsymbol{x}} \sum_i a_i (x_i - x_i')^2.$$

This toy model highlights neural networks' ability to encode numerous features representing real-world concepts into a compressed representation, providing insights into the superposition phenomenon observed in neural networks trained on real data.

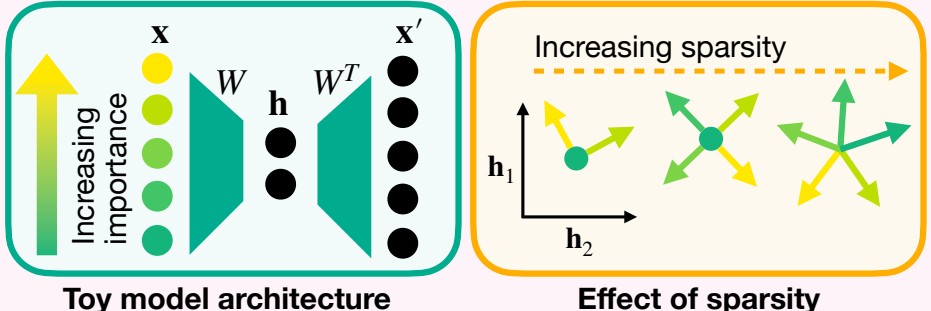

Figure 5: Illustration of the toy model architecture and the effects of sparsity. (left) Transformation of a five-feature input vector $\boldsymbol{x}$ into a two-dimensional hidden representation $\boldsymbol{h}$, and its reconstruction as $\boldsymbol{x}'$ using the weight matrix $\boldsymbol{W}$ and its transpose, with feature importance indicated by a color gradient from yellow to green. (right) The effect of increasing feature sparsity $S$ on the encoding capacity of the network, highlighting the network's enhanced ability to represent features in superposition as sparsity increases from 0 to 0.9, illustrated by arrows in the activation space $\boldsymbol{h}$, which correspond to the columns of the matrix $\boldsymbol{W}$.

Research on superposition, including works by (Elhage et al., 2022b; Scherlis et al., 2023; Henighan et al., 2023), often investigates simplified models. However, understanding superposition in practical, transformer-based scenarios is crucial for real-world applications, as pioneered by Gurnee et al. (2023).

The need for understanding networks despite polysemanticity has led to various approaches: One involves training models without superposition (Jermyn et al., 2022), for example, using a softmax linear unit (Elhage et al., 2022a) as an activation function to empirically increase the number of *monosemantic* neurons, but at the cost of making other neurons less interpretable. From a capabilities standpoint, polysemanticity may be desirable as it allows models to represent more concepts with limited compute, making training cheaper. Overall, engineering monosemanticity has proven challenging (Bricken et al., 2023) and may be impractical until we have orders of magnitude more compute available.

Another approach is to train networks in a standard way (creating polysemanticity) and use post-hoc analysis to find the feature directions in activation space, for example, with Sparse Autoencoders (SAEs). SAEs aim to find the true, disentangled features in an uncompressed representation by learning a sparse overcomplete basis that describes the activation space of the trained model (Bricken et al., 2023; Sharkey et al., 2022b; Cunningham et al., 2024) (also see Section 4.2).

**If not neurons, what are features then?**  We want to identify the fundamental units of neural networks, which we call *features*. Initially, neurons seemed likely candidates. However, this view fell short, particularly in transformer models where neurons often represent multiple concepts, a phenomenon known as polysemanticity. The superposition hypothesis addresses this, proposing that due to limited representational capacity, neural networks compress numerous features into the confined space of neurons, complicating interpretation.

This raises the question: *How are features encoded if not in discrete neuron units?* While a priori features could be encoded in an arbitrarily complex, non-linear structure, a growing body of theoretical arguments and empirical evidence supports the hypothesis that features are commonly represented linearly, *i.e.*, as linear combinations of neurons – hence, as directions in representation space. This perspective promises to enhance our comprehension of neural networks by providing a more interpretable and manipulable framework for their internal representations.

> **Hypothesis 2: Linear Representation**
>
> Features are directions in activation space, *i.e.*, linear combinations of neurons.

The *linear representation* hypothesis suggests that neural networks frequently represent high-level features as linear directions in activation space. This hypothesis can simplify the understanding and manipulation of neural network representations (Nanda et al., 2023b). The prevalence of linear layers in neural network architectures favors linear representations. Matrix multiplication in these layers most readily processes linear features, while more complex non-linear encodings would require multiple layers to decode.

However, recent work by Engels et al. (2024) provides evidence against a strict formulation of the linear representation hypothesis by identifying circular features representing days of the week and months of the year. These multi-dimensional, non-linear representations were shown to be used for solving modular arithmetic problems in days and months. Intervention experiments confirmed that these circular features are the fundamental unit of computation in these tasks, and the authors developed methods to decompose the hidden states, revealing the circular representations.

Establishing non-linearity can be challenging. For example, Li et al. (2023a) initially found that in a GPT model trained on Othello, the board state could only be decoded with a non-linear probe when represented in terms of white and black pieces, seemingly violating the linearity assumption. However, Nanda (2023c); Nanda et al. (2023b) later showed that a linear probe sufficed when the board state was decoded in terms of "one's own" and "the opponent's" pieces, reaffirming the linear representation hypothesis in this case. In contrast, the work by Engels et al. (2024) provides a clear and convincing existence proof for non-linear, multi-dimensional representations in language models.

While the linear representation hypothesis remains a useful simplification, it is important to recognize its limitations and the potential role of non-linear representations (Sharkey et al., 2022a). As neural networks continue to evolve, ongoing reevaluation of the hypothesis is crucial, particularly considering the possible emergence of non-linear features under optimization pressure for interpretability (Hubinger, 2022). Alter-

native perspectives, such as the polytope lens proposed by Black et al. (2022), emphasize the impact of non-linear activation functions and discrete polytopes formed by piecewise linear activations as potential primitives of neural network representations.

Despite these exceptions, empirical evidence largely supports the linear representation hypothesis in many contexts, especially for feedforward networks with ReLU activations. Semantic vector calculus in word embeddings (Mikolov et al., 2013), successful linear probing (Alain & Bengio, 2016; Belinkov, 2021), sparse dictionary learning (Bricken et al., 2023; Cunningham et al., 2024; Deng et al., 2023), and linear decoding of concepts (O'Mahony et al., 2023), tasks (Hendel et al., 2023), functions (Todd et al., 2023), sentiment (Tigges et al., 2024), refusal (Arditi et al., 2024), and relations (Hernandez et al., 2023; Chanin et al., 2023) in large language models all point to the prevalence of linear representations. Moreover, linear addition techniques for model steering (Turner et al., 2023; Sakarvadia et al., 2023a; Li et al., 2023b) and *representation engineering* (Zou et al., 2023) highlight the practical implications of linear feature representations.

Building upon the linear representation hypothesis, recent work investigated the structural organization of these linear features within activation space. Park et al. (2024) reveal a geometric framework for categorical and hierarchical concepts in large language models. Their findings demonstrate that simple categorical concepts (*e.g.*, mammal, bird) are represented as simplices in the activation space, while hierarchically related concepts are orthogonal. This geometric analysis aligns with earlier observations on feature clustering and splitting in neural networks (Elhage et al., 2022b). It suggests that the linear features are not merely scattered directions but are organized to reflect semantic relationships and hierarchies.

### 3.3 Circuits as Computational Primitives and Motifs as Universal Circuit Patterns

Having defined features as directions in activation space as the fundamental units of neural network representation, we now explore their computation. Neural networks can be conceptualized as computational graphs, within which *circuits* are sub-graphs consisting of linked features and the weights connecting them. Similar to how features are the representational primitive, circuits function as the computational primitive (Michaud et al., 2023) and the primary building block of these networks (Olah et al., 2020).

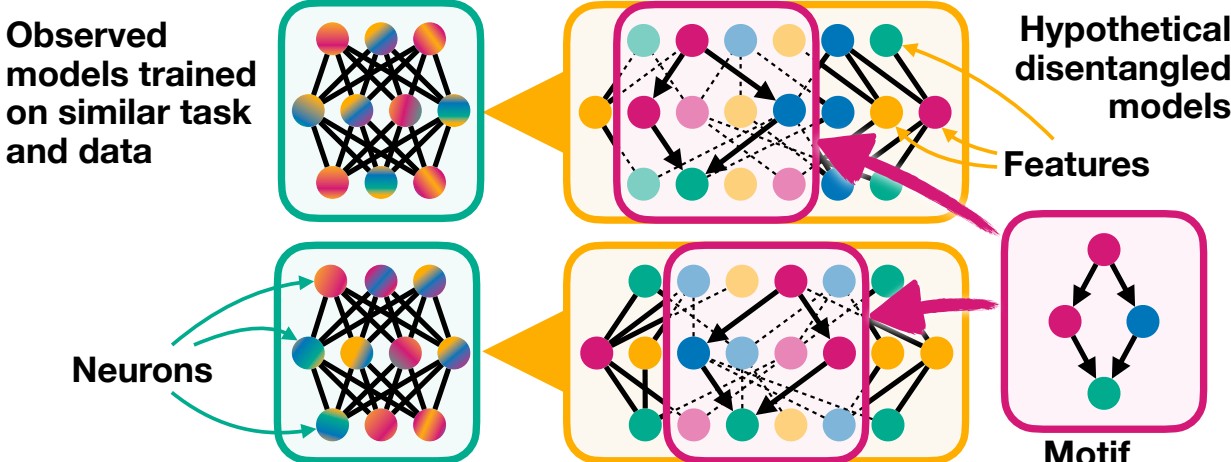

Figure 6: Comparing observed models (left) and corresponding hypothetical *disentangled* models (right) trained on similar tasks and data. The observed models show different neuronal activation patterns, while the dissection into feature-level *circuits* reveals a *motif* - a shared circuit pattern emerging across models, hinting at *universality* – models converging on similar solutions based on common underlying principles.

---

**Definition 3: Circuit**

Circuits are sub-graphs of the network, consisting of features and the weights connecting them.

---

The decomposition of neural networks into circuits for interpretability has shown significant promise, particularly in small models trained for specific tasks such as addition, as seen in the work of Nanda et al. (2023a) and Quirke & Barez (2023). Scaling such a *comprehensive* circuit analysis to broader behaviors in large language models remains challenging. However, there has been notable progress in scaling circuit analysis of *narrow behaviors* to larger circuits and models, such as indirect object identification (Wang et al., 2023) and greater-than computations (Hanna et al., 2023) in GPT-2 and multiple-choice question answering (Lieberum et al., 2023).

In search of general and universal circuits, researchers focus particularly on more general and transferable behaviors. McDougall et al. (2023)'s work on copy suppression in GPT-2's attention heads sheds light on model calibration and self-repair mechanisms. Davies et al. (2023) and Feng & Steinhardt (2023) focus on how large language models represent symbolic knowledge through variable binding and entity-attribute binding, respectively. Yu et al. (2023); Nanda et al. (2023c); Lv et al. (2024); Chughtai et al. (2024); Ortu et al. (2024) explore mechanisms for factual recall, revealing how circuits dynamically balance pre-trained knowledge with new contextual information. Lan & Barez (2023) extend circuit analysis to sequence continuation tasks, identifying shared computational structures across semantically related sequences.

More promisingly, some repeating patterns have shown *universality* across models and tasks. These universal patterns are called *motifs* (Olah et al., 2020) and can manifest not just as specific circuits or features but also as higher-level behaviors emerging from the interaction of multiple components. Examples include the curve detectors found across vision models (Cammarata et al., 2021; 2020), induction circuits enabling in-context learning (Olsson et al., 2022), and the phenomenon of branch specialization in neural networks (Voss et al., 2021). Motifs may also capture how models leverage tokens for working memory or parallelize computations in a divide-and-conquer fashion across representations. The significance of motifs lies in revealing the common structures, mechanisms, and strategies that naturally emerge across neural architectures, shedding light on the fundamental building blocks underlying their intelligence. Figure 6 contrasts observed neural network models with hypothetical disentangled models, illustrating how a shared circuit pattern can emerge across different models trained on similar tasks and data, hinting at an underlying *universality*.

> **Definition 4: Motif**
>
> Motifs are repeated patterns within a network, encompassing either features or circuits that emerge across different models and tasks.

**Universality Hypothesis.** Following the evidence for motifs, we can propose two versions for a *universality* hypothesis regarding the convergence of features and circuits across neural network models:

> **Hypothesis 3: Weak Universality**
>
> There are underlying principles governing how neural networks learn to solve certain tasks. Models will generally converge on analogous solutions that adhere to the common underlying principles. However, the specific *features* and *circuits* that implement these principles can vary across different models based on factors like hyperparameters, random seeds, and architectural choices.

> **Hypothesis 4: Strong Universality**
>
> The *same* core features and circuits will universally and consistently arise across all neural network models trained on similar tasks and data distributions and using similar techniques, reflecting a set of fundamental computational *motifs* that neural networks inherently gravitate towards when learning.

The universality hypothesis posits a convergence in forming features and circuits across various models and tasks, which could significantly ease interpretability efforts in AI. It proposes that artificial and biological

neural networks share similar features and circuits, suggesting a standard underlying structure (Chan et al., 2023; Sucholutsky et al., 2023; Kornblith et al., 2019). This idea posits that there is a fundamental basis in how neural networks, irrespective of their specific configurations, process and comprehend information. This could be due to inbuilt inductive biases in neural networks or *natural abstractions* (Chan et al., 2023) – concepts favored by the natural world that any cognitive system would naturally gravitate towards.

Evidence for this hypothesis comes from *cross-species neural structures* in neuroscience, where similar neural structures and functions are found in different species (Kirchner, 2023). Additionally, machine learning models, including neural networks, tend to converge on similar features, representations, and classifications across different tasks and architectures (Chen et al., 2023a; Hacohen et al., 2020; Li et al., 2015; Bricken et al., 2023). Marchetti et al. (2023) provide mathematical support for emerging universal features.

While various studies support the universality hypothesis, questions remain about the extent of feature and circuit similarity across different models and tasks. In the context of mechanistic interpretability, this hypothesis has been investigated for neurons (Gurnee et al., 2024), group composition circuits (Chughtai et al., 2023), and modular task processing (Variengien & Winsor, 2023), with evidence for the weak but not the strong formulation (Chughtai et al., 2023).

### 3.4 Emergence of World Models and Simulated Agents

**Internal World Models.** World models are internal causal models of an environment formed within neural networks. Traditionally linked with reinforcement learning, these models are *explicitly* trained to develop a compressed spatial and temporal representation of the training environment, enhancing downstream task performance and sample efficiency through training on internal hallucinations (Ha & Schmidhuber, 2018). However, in the context of our survey, our focus shifts to *internal world models* that potentially form *implicitly* as a by-product of the training process, especially in LLMs trained on next-token prediction – also called GPT.

LLMs are sometimes characterized as *stochastic parrots* (Bender et al., 2021). This label stems from their fundamental operational mechanism of predicting the next word in a sequence, which is seen as relying heavily on memorization. From this viewpoint, LLMs are thought to form complex correlations based on observational data but cannot develop causal models of the world due to their lack of access to interventional data (Pearl, 2009).

An alternative perspective on LLMs comes from the *active inference* framework (Salvatori et al., 2023), a theory rooted in cognitive science and neuroscience. Active inference postulates that the objective of minimizing prediction error, given enough representative capacity, is adequate for a learning system to develop complex world representations, behaviors, and abstractions. Since language inherently mirrors the world, these models could implicitly construct linguistic and broader world models (Kulveit et al., 2023).

The *simulation* hypothesis suggests that models designed for prediction, such as LLMs, will eventually simulate the causal processes underlying data creation. Seen as an extension of their drive for efficient compression, this hypothesis implies that adequately trained models like GPT could develop *internal world models* as a natural outcome of their predictive training (janus, 2022; Shanahan et al., 2023).

> **Hypothesis 5: Simulation**
>
> A model whose objective is text prediction will simulate the causal processes underlying the text creation if optimized sufficiently strongly (janus, 2022).

In addition to theoretical considerations for emergent causal world models (Richens & Everitt, 2024; Nichani et al., 2024), mechanistic interpretability is starting to provide empirical evidence on the types of internal world models that may emerge in LLMs. The ability to internally represent the board state in games like chess (Karvonen, 2024) or Othello (Li et al., 2023a; Nanda et al., 2023b), create linear abstractions of spatial and temporal data (Gurnee & Tegmark, 2024), and structure complex representations of mazes, demonstrating an understanding of maze topology and pathways (Ivanitskiy et al., 2023) highlight the growing abstraction

capabilities of LLMs. Li et al. (2021) identified contextual word representations that function as models of entities and situations evolving throughout a discourse, akin to linguistic models of dynamic semantics. Patel & Pavlick (2022) demonstrated that LLMs can map conceptual domains (e.g, direction, color) to grounded world representations given a few examples, suggesting they learn rich conceptual spaces (Gardenfors, 2004) reflective of the non-linguistic world.

The *prediction orthogonality* hypothesis further expands on this idea: It posits that prediction-focused models like GPT may simulate agents with various objectives and levels of optimality. In this context, GPT are simulators, simulating entities known as *simulacra* that can be either agentic or non-agentic, with different objectives from the simulator itself (janus, 2022; Shanahan et al., 2023). The implications of the simulation and prediction orthogonality hypotheses for AI safety and alignment are discussed in Section 6.

> **Hypothesis 6: Prediction Orthogonality**
>
> A model whose objective is prediction can simulate agents who optimize toward any objectives with any degree of optimality (janus, 2022).

In conclusion, the evolution of LLMs from simple predictive models to entities potentially possessing complex *internal world models*, as suggested by the *simulation* hypothesis and supported by mechanistic interpretability studies, represents a significant shift in our understanding of these systems. This evolution challenges us to reconsider LLMs' capabilities and future trajectories in the broader landscape of AI development.

## 4 Core Methods

Mechanistic interpretability (MI) employs various tools, from observational analysis to causal interventions. This section provides a comprehensive overview of these methods, beginning with a taxonomy that categorizes approaches based on their key characteristics (Section 4.1). We then survey observational (Section 4.2), followed by interventional techniques (Section 4.3). Finally, we study their synergistic interplay (Section 4.4). Figure 7 offers a visual summary of the methods and techniques unique to mechanistic interpretability.

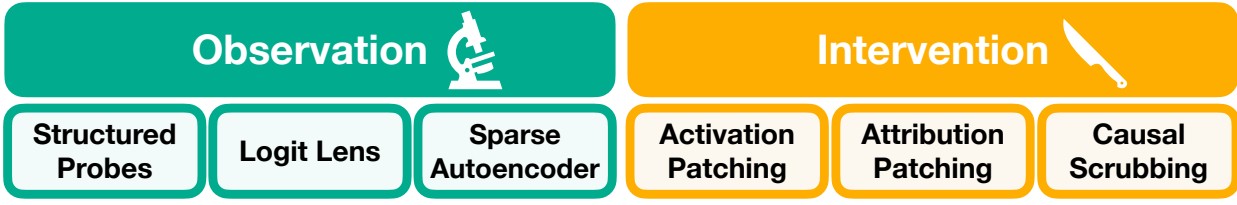

Figure 7: Overview of key methods and techniques in mechanistic interpretability research. Observational approaches include structured probes, logit lens variants, and sparse autoencoders (SAEs). Interventional methods, focusing on causal understanding, encompass activation patching variants for uncovering causal mechanisms and causal scrubbing for hypothesis evaluation.

### 4.1 Taxonomy of Mechanistic Interpretability Methods

We propose a taxonomy based on four key dimensions: causal nature, learning phase, locality, and comprehensiveness (Table 1).

The causal nature of methods ranges from purely observational, which analyze existing representations without direct manipulation, to interventional approaches that actively perturb model components to establish causal relationships. The learning phase dimension distinguishes between post-hoc techniques applied to trained models and intrinsic methods that enhance interpretability during the training process itself.

Locality refers to the scope of analysis, spanning from individual neurons (*e.g.*, feature visualization) to entire model architectures (*e.g.*, causal abstraction). Comprehensiveness varies from partial insights into specific components to holistic explanations of model behavior.

Table 1: Taxonomy of Mechanistic Interpretability Methods

| Method | Causal Nature | Phase | Locality | Comprehensiveness | Key Examples |
|---|---|---|---|---|---|
| Feature Visualization | Observation | Post-hoc | Local | Partial | Zeiler & Fergus (2014) Zimmermann et al. (2021) |
| Exemplar methods | Observation | Post-hoc | Local | Partial | Grosse et al. (2023) Garde et al. (2023) |
| Probing Techniques | Observation | Post-hoc | Both | Both | McGrath et al. (2022) Gurnee et al. (2023) |
| Structured Probes | Observation | Post-hoc | Both | Both | Burns et al. (2023) |
| Logit Lens Variants | Observation | Post-hoc | Global | Partial | nostalgebraist (2020) Belrose et al. (2023) |
| Sparse Autoencoders | Observation | Post-hoc | Both | Comprehensive | Cunningham et al. (2024) Bricken et al. (2023) |
| Activation Patching | Intervention | Post-hoc | Local | Partial | Meng et al. (2022a) Wang et al. (2023) |
| Path Patching | Intervention | Post-hoc | Both | Both | Goldowsky-Dill et al. (2023) |
| Causal Abstraction | Intervention | Post-hoc | Global | Comprehensive | Geiger et al. (2023a) Geiger et al. (2023b) Wu et al. (2023a) |
| Hypothesis Testing | Intervention | Post-hoc | Global | Comprehensive | Chan et al. (2022) Jenner et al. (2023) |
| Intrinsic Methods | – | Pre/During | Global | Comprehensive | Elhage et al. (2022a) Liu et al. (2023a) |

The categorization is based on the methods' general tendencies. Some methods can offer local and global or partial and comprehensive interpretability depending on the scope of the analysis and application. Probing techniques can range from local to global and partial to comprehensive; simple linear probes might offer local insights into individual *features*, while more sophisticated structured probes can uncover global patterns. Sparse autoencoders decompose individual neuron activations (local) but aim to disentangle features across the entire model (global). Path patching extends local interventions to global model understanding by tracing information flow across layers, demonstrating how local perturbations can yield broader insights.

In practice, mechanistic interpretability research involves both method development and their application. When applying methods to understand a model, combining techniques from multiple categories is often necessary and beneficial to build a more comprehensive understanding (Section 4.4).

## 4.2 Observation

Mechanistic interpretability draws from observational methods that analyze the inner workings of neural networks, with many of these methods preceding the field itself. For a detailed exploration of inner interpretability methods, refer to (Räuker et al., 2023). Two prominent categories are example-based methods and feature-based methods:

i.) **Exemplar methods** identify real input examples that highly activate specific neurons or layers. This helps pinpoint influential data points that maximize neuron activation within the neural network (Grosse et al., 2023; Garde et al., 2023; Nanfack et al., 2024).

ii.) **Feature visualization** encompasses techniques that generate synthetic inputs to optimize neuron activation. These visualizations reveal how neurons respond to stimuli and which features are sensitive to (Zeiler & Fergus, 2014; Zimmermann et al., 2021). By inspecting the synthetic inputs that drive neuron behavior, we can hypothesize about the features encoded by those neurons.

**Probing for Features.** Probing (Alain & Bengio, 2016; Hewitt & Manning, 2019) involves training a classifier using the activations of a model, with the classifier's performance subsequently observed to deduce insights about the model's behavior and internal representations. However, the probe's performance may often reflect its own learning capacities more than the actual characteristics of the model's representations (Belinkov, 2021). This dilemma has led researchers to investigate the ideal balance between the complexity of a probe and its capacity to accurately represent the model's features (Cao et al., 2021; Voita & Titov, 2020).

The *linear representation* hypothesis offers a resolution to this issue. Under this hypothesis, the failure of a simple linear probe to detect certain features suggests their absence in the model's representations. Conversely, suppose a more complex probe succeeds where a simpler one fails. In that case, it implies that the model contains features that a complex function can combine into the target feature but that the target feature itself is not explicitly represented. Thus, the hypothesis implies that using linear probes could suffice in most cases, circumventing the complexity considerations generally associated with probing (Belinkov, 2021).

McGrath et al. (2022) analyzed chess knowledge acquisition in AlphaZero, revealing the emergence of strategic concepts during training. In language models, Gurnee et al. (2023) introduced *sparse probing* to decode internal neuron activations to understand feature representation and sparsity. They show that early layers use sparse combinations of neurons to represent many features in superposition, while middle layers seem to have dedicated *monosemantic* neurons for higher-level contextual features.

Probing is limited in drawing causal or behavioral conclusions. Its primarily observational nature focuses on how information is encoded rather than how it is used (see Figure 1), necessitating careful analysis and integration with interventional techniques (Section 4.3), or alternative approaches (Elazar et al., 2021). While in explainable AI (Linardatos et al., 2020), probing has primarily analyzed high-level concepts like linguistic representations (Tenney et al., 2019; Dalvi et al., 2019), MI aims to probe towards uncovering underlying computational processes and functionality. This shift in goals towards uncovering mechanistic computation is a nuanced distinction rather than a clear-cut line between probing in MI and the broader explainability field.

**Structured Probes.** While focusing on bottom-up, mechanistic interpretability approaches, we can also consider integrating top-down, concept-based structured probes with mechanistic interpretability.

Structured probes aid conceptual interpretability, probing language models for complex features like truth representations. Notably, Burns et al. (2023)'s *contrast-consistent search* identifies linear projections exhibiting logical consistency in hidden states, contrasting truth values for statements and negations.

However, structured probes face significant challenges in unsupervised probing scenarios. As Farquhar et al. (2023) showed, arbitrary features, not just knowledge-related ones, can satisfy contrast consistency equally well, raising doubts about scalability. For example, the loss may capture *simulation* of knowledge from hypothesized *simulacra* within sufficiently powerful language models rather than the models' true knowledge. Furthermore, Farquhar et al. (2023) demonstrates self-supervised probing methods (like (Burns et al., 2023)) often detect prominent but unintended distractor features in the data. The discovered features are also highly sensitive to prompt choice, and there is no principled way to select prompts that would reliably surface a model's true knowledge.

While structured probes primarily focus on high-level conceptual representations (Zou et al., 2023), their findings could potentially inform or complement mechanistic interpretability efforts. For instance, identifying truth directions through structured probes could help guide targeted interventions or analyze the underlying circuits responsible for truthful behavior using mechanistic techniques such as activation patching or circuit tracing (Section 4.3). Conversely, mechanistic methods could provide insights into how truth representations emerge and are computed within the model, addressing some of the challenges faced by unsupervised structured probes.

**Logit Lens.** The *logit lens* (nostalgebraist, 2020) provides a window into the model's predictive process by applying the final classification layer (which projects the residual stream activation into logits/vocabulary

space) to intermediate activations of the residual stream, revealing how prediction confidence evolves across computational stages. This is possible because transformers tend to build their predictions across layers iteratively (Geva et al., 2022). Extensions of this approach include the tuned lens (Belrose et al., 2023), which trains affine probes to decode hidden states into probability distributions over the vocabulary, and the Future Lens (Pal et al., 2023), which explores the extent to which individual hidden states encode information about subsequent tokens.

Researchers have also investigated techniques that bypass intermediate computations to probe representations directly. Din et al. (2023) propose using linear transformations to approximate hidden states from different layers, revealing that language models often predict final outputs in early layers. Dar et al. (2022) present a theoretical framework for interpreting transformer parameters by projecting them into the embedding space, enabling model alignment and parameter transfer across architectures.

Other techniques focus on interpreting specific model components or submodules. The DecoderLens (Langedijk et al., 2023) allows analyzing encoder-decoder transformers by cross-attending intermediate encoder representations in the decoder, shedding light on the information flow within the encoder. The Attention Lens (Sakarvadia et al., 2023b) aims to elucidate the specialized roles of attention heads by translating their outputs into vocabulary tokens via learned transformations.

**Feature Disentanglement via Sparse Dictionary Learning.** Recent work suggests that the essential elements in neural networks are linear combinations of neurons representing features in superposition (Elhage et al., 2022b). To disentangle these features, researchers have developed sparse autoencoders (SAEs), which decompose neural network activations into individual component features (Sharkey et al., 2022b; Cunningham et al., 2024). This process, known as sparse dictionary learning, reconstructs activation vectors as sparse linear combinations of directional vectors within the activation space (Olshausen & Field, 1997).

The theoretical foundations of SAEs are rooted in work on *disentangled* representations. Whittington et al. (2022) demonstrate that autoencoders can recover ground truth features under conditions of feature sparsity and non-negativity. Furthermore, Garfinkle & Hillar (2019) provides guarantees for the uniqueness and stability of dictionaries for sparse representation, even in the presence of noise. These theoretical underpinnings support SAEs' ability to uncover true, disentangled features underlying the data distribution.

In practice, SAEs stand out for their simplicity and scalability (Sharkey et al., 2022b). They incorporate sparsity regularization to encourage learning sparse yet meaningful data representations, with the precise tuning of the sparsity penalty on hidden activations critical in dictating the autoencoder's sparsity level. We provide an overview of the SAE architecture in Figure 8.

SAEs' dictionary features exhibit higher scores on autointerpretability metrics and increased monosemanticity (Bricken et al., 2023; Cunningham et al., 2024; Sharkey et al., 2022b). They are scalable to state-of-the-art models and can detect safety-relevant features (Templeton et al., 2024), measure feature sparsity (Deng et al., 2023), and interpret reward models in reinforcement learning-based language models (Marks et al., 2023a).

Evaluating SAE quality remains challenging due to the lack of ground-truth interpretable features. Researchers have addressed this through various approaches: Karvonen et al. (2024) proposed using language models trained on chess and Othello transcripts as testbeds, providing natural collections of interpretable features. Sharkey et al. (2022b) constructed a toy model with traceable features, while Makelov et al. (2024); Makelov (2024) compared SAE results with supervised features in large language models to demonstrate their viability.

The versatility of SAEs extends to various neural network architectures. They have been successfully applied to transformer attention layers (Kissane et al., 2024) and convolutional neural networks (Gorton, 2024). Notably, Gorton (2024) applied SAEs to the early vision layers of InceptionV1, uncovering new interpretable features, including additional curve detectors not apparent from examining individual neurons (Cammarata et al., 2020).

In circuit discovery, SAEs have shown particular promise (see also Section 4.4). He et al. (2024) proposed a circuit discovery framework alternative to activation patching (discussed in Section 4.3.1), leveraging

dictionary features decomposed from all modules writing to the residual stream. Similarly, O'Neill & Bui (2024) employed discrete sparse autoencoders for discovering interpretable circuits in large language models.

Recent advancements have focused on improving SAE performance and addressing limitations. Rajamanoharan et al. (2024) introduced a gating mechanism to separate the functionalities of determining which directions to use and estimating their magnitudes, mitigating shrinkage – the systematic underestimation of feature activations. An alternative approach by Dunefsky et al. (2024) uses transcoders to faithfully approximate a densely activating MLP layer with a wider, sparsely-activating MLP layer, offering another path to interpretable feature discovery, a type of sparse distillation (slavachalnev, 2024).

---

### Sparse Dictionary Learning

Sparse autoencoders (Cunningham et al., 2024) are proposed as a solution to *polysemantic* neurons. The problem of *superposition* is mathematically formalized as *sparse dictionary learning* (Olshausen & Field, 1997) problem to decompose neural network activations into *disentangled* component features. The goal is to learn a dictionary of vectors $\{\boldsymbol{f}_k\}_{k=1}^{n_{\text{feat}}} \subset \mathbb{R}^d$ that can represent the unknown, ground truth network features as sparse linear combinations. If successful, the learned dictionary contains *monosemantic* neurons corresponding to *features* (Sharkey et al., 2022b). The autoencoder architecture consists of an encoder and a ReLU activation function, expanding the input dimensionality to $d_{\text{hid}} > d_{\text{in}}$. The encoder's output is given by:

$$\boldsymbol{h} = \text{ReLU}(\boldsymbol{W}_{\text{enc}}\boldsymbol{x} + \boldsymbol{b}), \tag{1}$$

$$\boldsymbol{x}' = \boldsymbol{W}_{\text{dec}}\boldsymbol{h} = \sum_{i=0}^{d_{\text{hid}}-1} h_i \boldsymbol{f}_i, \tag{2}$$

where $\boldsymbol{W}_{\text{enc}}, \boldsymbol{W}_{\text{dec}}^{\top} \in \mathbb{R}^{d_{\text{hid}} \times d_{\text{in}}}$ and $\boldsymbol{b} \in \mathbb{R}^{d_{\text{hid}}}$. The parameter matrix $\boldsymbol{W}_{\text{dec}}$ forms the feature dictionary, with rows $\boldsymbol{f}_i$ as dictionary features. The autoencoder is trained to minimize the loss, where the $L^1$ penalty on $\boldsymbol{h}$ encourages sparse reconstructions using the dictionary features,

$$\mathcal{L}(\boldsymbol{x}) = |\boldsymbol{x} - \boldsymbol{x}'|_2^2 + \alpha|\boldsymbol{h}|_1. \tag{3}$$

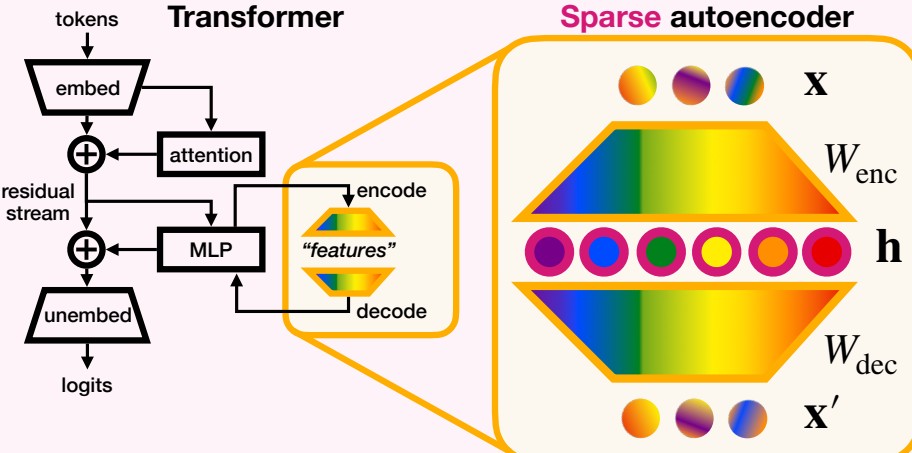

Figure 8: Illustration of a sparse autoencoder applied to the MLP layer activations, consisting of an encoder that increases dimensionality while emphasizing sparse representations and a decoder that reconstructs the original activations using the learned feature dictionary.

## 4.3 Intervention

**Causality as a Theoretical Foundation.** The theory of causality (Pearl, 2009) provides a mathematically precise framework for mechanistic interpretability, offering a rigorous approach to understanding high-level semantics in neural representations (Geiger et al., 2023a). By treating neural networks as causal models, with their *compute graphs serving as causal graphs*, researchers can perform precise interventions and examine the roles of individual parameters (Mueller et al., 2024). This causal perspective on interpretability has led to the development of various intervention techniques, including activation patching (Section 4.3.1), causal abstraction (Section 4.3.2), and hypothesis testing methods (Section 4.3.3).

### 4.3.1 Activation Patching

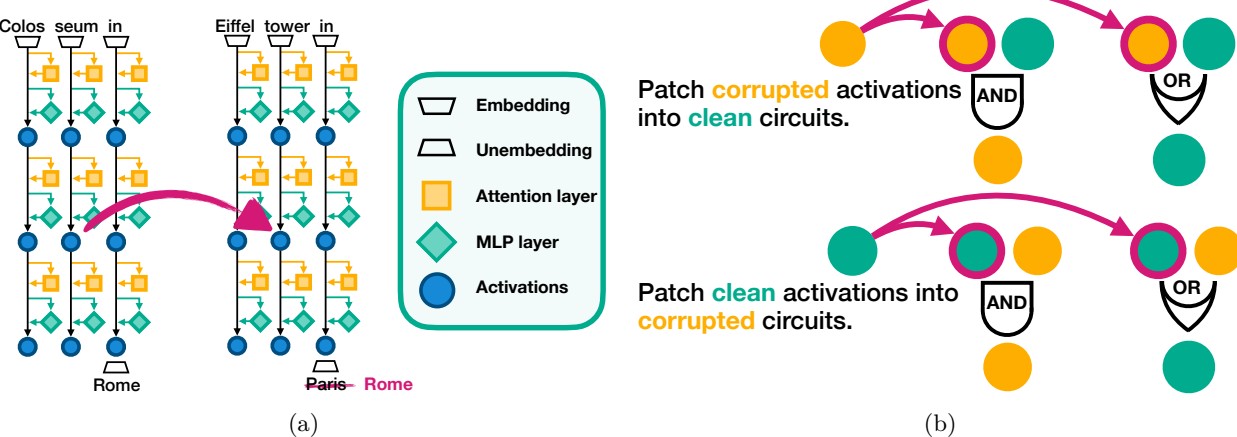

Figure 9: (a) Activation patching in a transformer model. Left: The model processes the clean input "Colosseum in Rome," caching the latent activations (step *i.*). Right: The model runs with the corrupted input "Eiffel Tower in Paris" (step *ii.*). The pink arrow shows an MLP layer activation (green diamond) patched from the clean run into the corrupted run (step *iii.*). This causes the prediction to change from "Paris" to "Rome," demonstrating how the significance of the patched component is determined (step *iv.*). By comparing these carefully selected inputs, researchers can control for confounding circuitry and isolate the specific circuit responsible for the location prediction behavior. (b) Activation patching directions: Top: Patching corrupted activations (orange) into clean circuits (turquoise) reveals *sufficient* components for identifying OR logic scenarios. Bottom: Patching clean activations (green) into corrupted circuits (orange) reveals *necessary* components that are useful for identifying AND logic scenarios. The AND and OR gates demonstrate how these patching directions uncover different logical relationships between model components.

Activation patching is a collective term for a set of causal intervention techniques that manipulate neural network activations to shed light on the decision-making processes within the model. These techniques, including causal tracing (Meng et al., 2022a), interchange intervention (Geiger et al., 2021b), causal mediation analysis (Vig et al., 2020), and causal ablation (Wang et al., 2023), share the common goal of modifying a neural model's internal state by replacing specific activations with alternative values, such as zeros, mean activations across samples, random noise, or activations from a different forward pass (Figure 9a).

The primary objective of activation patching is to isolate and understand the role of specific components or circuits within the model by observing how changes in activations affect the model's output. This enables researchers to infer the function and importance of those components. Key applications include localizing behavior by identifying critical activations, such as understanding the storage and processing of factual information (Meng et al., 2022a; Geva et al., 2023; Goldowsky-Dill et al., 2023; Stolfo et al., 2023), and analyzing component interactions through circuit analysis to identify sub-networks within a model's computation graph that implement specified behaviors (Wang et al., 2023; Hanna et al., 2023; Lieberum et al., 2023; Hendel et al., 2023; Geva et al., 2023).

The standard protocol for activation patching (Figure 9a) involves:

step *i.* Running the model with a clean input and caching the latent activations;

step *ii.* Executing the model with a corrupted input;

step *iii.* Re-running the model with the corrupted input but substituting specific activations with those from the clean cache; and

step *iv.* Determining significance by observing the variations in the model's output during the third step, thereby highlighting the importance of the replaced components.

This process relies on comparing pairs of inputs: a clean input, which triggers the desired behavior, and a corrupted input, which is identical to the clean one except for critical differences that prevent the behavior. By carefully selecting these inputs, researchers can *control for confounding circuitry* and isolate the specific circuit responsible for the behavior.

Differences in patching direction – clean to corrupted (causal tracing) versus corrupted to clean (resample ablation) – provide insights into the sufficiency or necessity of model components for a given behavior. Clean to corrupted patching identifies activations sufficient for restoring clean performance, even if they are unnecessary due to redundancy, which is particularly informative in OR logic scenarios (Figure 9b, OR gate). Conversely, corrupted to clean patching determines the necessary activations for clean performance, which is useful in AND logic scenarios (Figure 9b, AND gate).

Activation patching can employ corruption methods, including zero-, mean-, random-, or resample ablation, each modulating the model's internal state in distinct ways. Resample ablation stands out for its effectiveness in maintaining consistent model behavior by not changing the data distribution too much (Zhang & Nanda, 2023). However, it is essential to be careful when interpreting the patching results, as breaking behavior by taking the model off-distribution is uninteresting for finding the relevant circuit (Nanda, 2023e).

**Path Patching and Subspace Activation Patching.** Path patching extends the activation patching approach to multiple edges in the computational graph (Wang et al., 2023; Goldowsky-Dill et al., 2023), allowing for a more fine-grained analysis of component interactions. For example, path patching can be used to estimate the direct and indirect effects of attention heads on the output logits. Subspace activation patching, also known as distributed interchange interventions (Geiger et al., 2023b), aims to intervene only on linear subspaces of the representation space where *features* are hypothesized to be encoded, providing a tool for more targeted interventions.

Recently, Ghandeharioun et al. (2024) introduced *patchscopes*, a framework that unifies and extends activation patching techniques: using the model's text generation to explain internal representations, it enables more flexible interventions across various interpretability tasks, improving early layer inspection and allowing for cross-model analysis.

**Limitations and Advancements.** Activation patching has several limitations, including the effort required to design input templates and counterfactual datasets, the need for human inspection to isolate important subgraphs, and potential second-order effects that can complicate the interpretation of results (Lange et al., 2023) and the *hydra effect* (McGrath et al., 2023; Rushing & Nanda, 2024) (see discussion in Section 7.2). Recent advancements aim to address these limitations, such as automated circuit discovery algorithms (Conmy et al., 2023), gradient-based methods for scalable component importance estimation like attribution patching (Nanda, 2023d; Syed et al., 2023), and techniques to mitigate self-repair interference during analysis (Ferrando & Voita, 2024).

### 4.3.2 Causal Abstraction

Causal abstraction (Geiger et al., 2021a; 2023a) provides a mathematical framework for mechanistic interpretability, treating neural networks and their explanations as causal models. This approach validates

explanations through interchange interventions on network activations (Jenner et al., 2023), unifying various interpretability methods such as LIME (Ribeiro et al., 2016), causal effect estimation (Feder et al., 2021), causal mediation analysis (Vig et al., 2020), iterated nullspace projection (Ravfogel et al., 2020), and circuit-based explanations (Geiger et al., 2023a).

To overcome computational limitations, *distributed alignment search* (Geiger et al., 2023b) introduced gradient-based distributed interchange interventions, extending causal abstraction to larger models (Wu et al., 2023b). Further advancements include *causal proxy models* (Wu et al., 2023a), which address the challenge of counterfactual observations.

Applications of causal abstraction span from linguistic phenomena analysis (Arora et al., 2024; Wu et al., 2022b), and evaluation of interpretability methods (Huang et al., 2024), to improving performance through representation finetuning (Wu et al., 2024), and improving efficiency via model distillation (Wu et al., 2022b).

### 4.3.3 Hypothesis Testing

In addition to the causal abstraction framework, several methods have been developed for rigorous hypothesis testing about neural network behavior. These methods aim to formalize and empirically validate explanations of how neural networks implement specific behaviors.

*Causal scrubbing* (Chan et al., 2022) formalizes hypotheses as a tuple $(\mathcal{G}, \mathcal{I}, c)$, where $\mathcal{G}$ is the model's computational graph, $\mathcal{I}$ is an interpretable computational graph hypothesized to explain the behavior, and $c$ maps nodes of $\mathcal{I}$ to nodes of $\mathcal{G}$. This method replaces activations in $\mathcal{G}$ with others that should be equivalent according to the hypothesis, measuring performance on the scrubbed model to validate the hypothesis.

*Locally consistent abstractions* (Jenner et al., 2023) offer a more permissive approach, checking the consistency between the neural network and the explanation only one step away from the intervention node. This method forms a middle ground between the strictness of full causal abstraction and the flexibility of causal scrubbing.

These methods form a hierarchy of strictness, with full causal abstractions being the most stringent, followed by locally consistent abstractions and causal scrubbing being the most permissive. This hierarchy highlights trade-offs in choosing stricter or more permissive notions, affecting the ability to find acceptable explanations, generalization, and mechanistic anomaly detection.

### 4.4 Integrating Observation and Intervention.

To comprehensively understand internal neural network mechanisms, combining observational and interventional methods is crucial. For instance, sparse autoencoders can be used to disentangle superposed features (Cunningham et al., 2024), followed by targeted activation patching to test the causal importance of these features (Wang et al., 2023). Similarly, the logit lens can track prediction formation across layers (nostalge-braist, 2020), with subsequent interventions confirming causal relationships at key points. Probing techniques can identify encoded information (Belinkov, 2021), which can then be subjected to causal abstraction (Geiger et al., 2023a) to understand how this information is utilized. This iterative refinement process, where broad observational methods guide targeted interventions and intervention results inform further observations, enables a multi-level analysis that builds a holistic understanding across different levels of abstraction. Recent work (Marks et al., 2024; Bushnaq et al., 2024; Braun et al., 2024; O'Neill & Bui, 2024; Ge et al., 2024) demonstrates the potential of integrating sparse autoencoders with automated circuits discovery (Conmy et al., 2023; Syed et al., 2023), combining feature-level analysis with circuit-level interventions to uncover the interplay between representation and mechanism.

## 5 Current Research

This section surveys current research in mechanistic interpretability across three approaches based on when and how the model is interpreted during training: Intrinsic interpretability methods are applied before training to enhance the model's inherent interpretability (Section 5.1). Developmental interpretability involves studying the model's learning dynamics and the emergence of internal structures during training (Section 5.2).

After training, post-hoc interpretability techniques are applied to gain insights into the model's behavior and decision-making processes (Section 5.3), including efforts towards uncovering general, transferable principles across models and tasks, as well as automating the discovery and interpretation of critical circuits in trained models (Section 5.4).

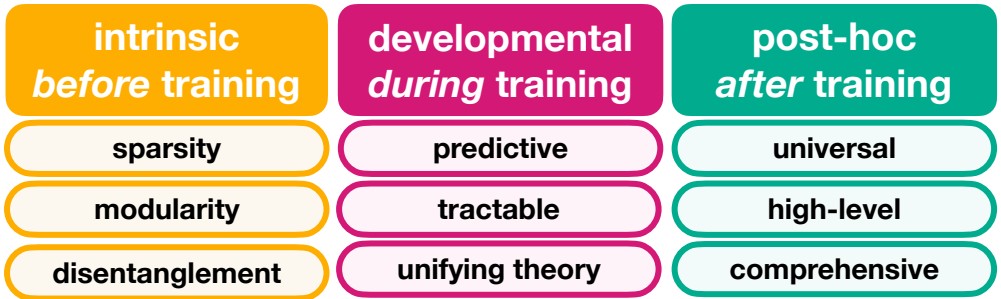

Figure 10: Key desiderata for interpretability approaches across training and analysis stages: (1) Intrinsic: Architectural biases for sparsity, *modularity*, and *disentangled* representations. (2) Developmental: Predictive capability for phase transitions, manageable number of critical transitions, and a unifying theory connecting observations to singularity geometry. (3) Post-hoc: Global, comprehensive, automated discovery of critical circuits, uncovering transferable principles across models/tasks, and extracting high-level causal mechanisms.

## 5.1 Intrinsic Interpretability

Intrinsic methods for mechanistic interpretability offer a promising approach to designing neural networks that are more amenable to *reverse engineering* without sacrificing performance. By encouraging *sparsity*, *modularity*, and *monosemanticity* through architectural choices and training procedures, these methods aim to make the reverse engineering process more tractable.

Intrinsic interpretability methods aim to constrain the training process to make learned programs more interpretable (Friedman et al., 2023b). This approach is closely related to neurosymbolic learning (Riegel et al., 2020) and can involve techniques like regularization with spatial structure, akin to the organization of information in the human brain (Liu et al., 2023a;b).

Recent work has explored various architectural choices and training procedures to improve the interpretability of neural networks. Jermyn et al. (2022) and Elhage et al. (2022a) demonstrate that architectural choices can affect monosemanticity, suggesting that models could be engineered to be more *monosemantic*. Sharkey (2023) propose using a bilinear layer instead of a linear layer to encourage monosemanticity in language models.

Liu et al. (2023a) and Liu et al. (2023b) introduce a biologically inspired spatial regularization regime called brain-inspired modular training for forming modules in networks during training. They showcase how this can help RNNs exhibit brain-like anatomical modularity without degrading performance, in contrast to naive attempts to use sparsity to reduce the cost of having more neurons per layer (Jermyn et al., 2022; Bricken et al., 2023).

Preceding the mechanistic interpretability literature, various works have explored techniques to improve interpretability, such as sparse attention (Zhang et al., 2021), adding $L^1$ penalties to neuron activations (Kasioumis et al., 2021; Georgiadis, 2019), and pruning neurons (Frankle & Carbin, 2019). These techniques have been shown to encourage sparsity, modularity, and disentanglement, which are essential aspects of intrinsic interpretability.

## 5.2 Developmental Interpretability

Developmental interpretability examines the learning dynamics and emergence of internal structures in neural networks over time, focusing on the formation of *features* and *circuits*. This approach complements static analyses by investigating critical phase transitions corresponding to significant changes in model behavior or

capabilities (Steinhardt, 2023; Schaeffer et al., 2023; Wei et al., 2022; Simon et al., 2023). While primarily a distinct field, developmental interpretability often intersects with mechanistic interpretability, as exemplified by Olsson et al. (2022)'s work. Their research, rooted in mechanistic interpretability, demonstrated how the emergence of in-context learning relates to specific training phase transitions, connecting microscopic changes (induction heads) with macroscopic observables (training loss).

A key motivation for developmental interpretability is investigating the *universality* of safety-critical patterns, aiming to understand how deeply ingrained and thereby resistant to safety fine-tuning capabilities like deception are. In addition, researchers hypothesize that emergent capabilities correspond to sudden circuit formation during training (Michaud et al., 2023), potentially allowing for prediction or control of their development.

Singular Learning Theory (SLT), developed by Watanabe (Watanabe, 2009; 2018), provides a rigorous framework for understanding overparameterized models' behavior and generalization. By quantifying model complexity through the *local learning coefficient*, SLT offers insights into learning phase transitions and the emergence of structure in the model (Lau et al., 2023). Recent work by Hoogland et al. (2024) applied this coefficient to identify developmental stages in transformer models, while Furman & Lau (2024) and Chen et al. (2023b) advanced SLT's scalability and application to the toy model of *superposition* (Figure 5), respectively.

While direct applications to phenomena such as generalization (Zhang et al., 2017), learning functions with increasing complexity (Nakkiran et al., 2019), and the transition from memorization to generalization (*grokking*) (Liu et al., 2022a; Power et al., 2022; Liu et al., 2022b; Nanda et al., 2023a; Varma et al., 2023; Thilak et al., 2022; Merrill et al., 2023; Liu et al., 2023c; Stander et al., 2023; Wang et al., 2024) are limited, these areas, along with neural scaling laws (Caballero et al., 2022; Liu & Tegmark, 2023; Michaud et al., 2023) (which can be connected to mechanistic insights (Hernandez et al., 2022)), represent promising future research directions.

In conclusion, developmental interpretability serves as an evolutionary theory lens for neural networks, offering insights into the emergence of structures and behaviors over time (Saphra, 2023). Drawing parallels from systems biology (Alon, 2019), this approach can apply concepts like network *motifs*, robustness, and *modularity* to neural network development, explaining how functional capabilities arise. Sometimes, understanding how structures came about is easier than analyzing the final product, similar to how biologists find certain features in organisms easier to explain in light of their evolutionary history. By studying the temporal aspects of neural network training, researchers can potentially uncover fundamental principles of learning and representation that may not be apparent from examining static, trained models alone.

## 5.3 Post-Hoc Interpretability

In applied mechanistic interpretability, researchers explore various facets and methodologies to uncover the inner workings of AI models. Some key distinctions are drawn between *global* versus *local* interpretability and *comprehensive* versus *partial* interpretability. Global interpretability aims to uncover general patterns and behaviors of a model, providing insights that apply broadly across many instances (Doshi-Velez & Kim, 2017; Nanda, 2023e). In contrast, local interpretability explains the reasons behind a model's decisions for particular instances, offering insights into individual predictions or behaviors. Comprehensive interpretability involves achieving a deep and exhaustive understanding of a model's behavior, providing a holistic view of its inner workings (Nanda, 2023e). In contrast, partial interpretability often applied to larger and more complex models, concentrates on interpreting specific aspects or subsets of the model's behavior, focusing on the application's most relevant or critical areas.

**Large Models – Narrow Behavior.** Circuit-style mechanistic interpretability aims to explain neural networks by *reverse engineering* the underlying mechanisms at the level of individual neurons or subgraphs. This approach assumes that neural vector representations encode high-level concepts and circuits defined by model weights encode meaningful algorithms (Olah et al., 2020; Cammarata et al., 2020). Studies on deep networks support these claims, identifying circuits responsible for detecting curved lines or object orientation (Cammarata et al., 2020; 2021; Voss et al., 2021).

This paradigm has been applied to language models to discover subnetworks (circuits) responsible for specific capabilities. Circuit analysis localizes and understands subgraphs within a model's computational graph responsible for specific behaviors. For large language models, this often involves narrow investigations into behaviors like multiple choice reasoning (Lieberum et al., 2023), indirect object identification (Wang et al., 2023), or computing operations (Hanna et al., 2023). Other examples include analyzing circuits for Python docstrings (Heimersheim & Jett, 2023), "an" vs "a" usage (Miller & Neo, 2023), and price tagging (Wu et al., 2023b). Case studies often construct datasets using templates filled by placeholder values to enable precise control for causal interventions (Wang et al., 2023; Hanna et al., 2023; Wu et al., 2023b).

**Toy Models – Comprehensive Analysis.** Small models trained on specialized mathematical or algorithmic tasks enable more comprehensive reverse engineering of learned algorithms (Nanda et al., 2023a; Zhong et al., 2023; Chughtai et al., 2023). Even simple arithmetic operations can involve complex strategies and multiple algorithmic solutions (Nanda et al., 2023a; Zhong et al., 2023). Characterizing these algorithms helps test hypotheses around generalizable mechanisms like variable binding (Feng & Steinhardt, 2023; Davies et al., 2023) and arithmetic reasoning (Stolfo et al., 2023). The work by Varma et al. (2023) builds on the work that analyzes transformers trained on modular addition (Nanda et al., 2023a) and explains *grokking* in terms of circuit efficiency, illustrating how a comprehensive understanding of a toy model can enable interesting analyses on top of that understanding.

**Towards Universality.** The ultimate goal is to uncover general principles that transfer across models and tasks, such as induction heads for in-context learning (Olsson et al., 2022), variable binding mechanisms (Feng & Steinhardt, 2023; Davies et al., 2023), arithmetic reasoning (Stolfo et al., 2023; Brinkmann et al., 2024), or retrieval tasks (Variengien & Winsor, 2023). Despite promising results, debates surround the *universality* hypothesis – the idea that different models learn similar features and circuits when trained on similar tasks. (Chughtai et al., 2023) finds mixed evidence for universality in group composition, suggesting that while families of circuits and features can be characterized, precise circuits and development order may be arbitrary.

**Towards High-level Mechanisms.** Causal interventions can extract a high-level understanding of computations and representations learned by large language models (Variengien & Winsor, 2023; Hendel et al., 2023; Feng & Steinhardt, 2023; Zou et al., 2023). Recent work focuses on intervening in internal representations to study high-level concepts and computations encoded. For example, Hendel et al. (2023) patched residual stream vectors to transfer task representations, while Feng & Steinhardt (2023) intervened on residual streams to argue that models generate IDs to bind entities to attributes. Techniques for *representation engineering* (Zou et al., 2023) extract reading vectors from model activations to stimulate or inhibit specific concepts. Although these interventions don't operate via specific mechanisms, they offer a promising approach for extracting high-level causal understanding and bridging bottom-up and top-down interpretability approaches.

## 5.4 Automation: Scaling Post-Hoc Interpretability

As models become more complex, automating key aspects of the interpretability workflow becomes increasingly crucial. Tracing a model's computational pathways is highly labor-intensive, quickly becoming infeasible as the model size increases. Automating the discovery of relevant circuits and their functional interpretation represents a pivotal step towards scalable and comprehensive model understanding (Nainani, 2024).

**Dissecting Models into Interpretable Circuits.** The first major automation challenge is identifying the critical computational sub-circuits or components underpinning a model's behavior for a given task. A pioneering line of work aims to achieve this via efficient **masking** or **patching** procedures. Methods like *automated circuit discovery* (Conmy et al., 2023) and *attribution patching* (Syed et al., 2023; Kramár et al., 2024) iteratively knock out model activations, pinpointing components whose removal has the most significant impact on performance. This masking approach has proven scalable even to large models (Lieberum et al., 2023).

Other techniques take a more top-down approach. Davies et al. (2023) specify high-level causal properties (desiderata) that components solving a target subtask should satisfy and then learn binary masks to expose those component subsets. Ferrando & Voita (2024) construct *information flow graphs* highlighting key nodes and operations by tracing attribution flows, enabling extraction of general information routing patterns across prediction domains.

Explicit architectural biases like modularity can further boost automation efficiency. Nainani (2024) find that models trained with *brain-inspired modular training* (Liu et al., 2023a) produce more readily identifiable circuits compared to standard training. Such domain-inspired inductive biases may prove increasingly vital as models grow more massive and monolithic.

**Interpreting Extracted Circuits.** Once critical circuit components have been isolated, the key remaining step is interpreting *what* computation those components perform. Sparse autoencoders are a prominent approach for interpreting extracted circuits by decomposing neural network activations into individual component *features*, as discussed in Section 4.2.

A novel paradigm uses large language models themselves as an interpretive tool. Bills et al. (2023) demonstrate generating natural language descriptions of individual neuron functions by prompting language models like GPT-4 to explain sets of inputs that activate a neuron. Mousi et al. (2023) similarly employ language models to annotate unsupervised neuron clusters identified via hierarchical clustering. Bai et al. (2024) describe the roles of neurons in vision networks with multimodal models. These methods can easily leverage more capable general-purpose models in the future. Foote et al. (2023) take a complementary graph-based approach in their neuron-to-graph tool: automatically extracting individual neurons' behavior patterns from training data as structured graphs amenable to visualization, programmatic comparisons, and property searches. Such representations could synergize with language model-based annotation to provide descriptions of neuron roles.

However, robustly interpreting the largest trillion-parameter models using automated techniques remains an open challenge. Another novel approach, mechanistic-interpretability-based program synthesis (Michaud et al., 2024), entirely sidesteps this complexity by auto-distilling the algorithm learned by a trained model into human-readable Python code without relying on further interpretability analyses or model architectural knowledge. As models become increasingly vast and opaque, such synergistic combinations of methods – uncovering circuits, annotating them, or altogether transcribing them into executable code – will likely prove crucial for maintaining insight and *oversight* when scaling model size.

## 6   Relevance to AI Safety

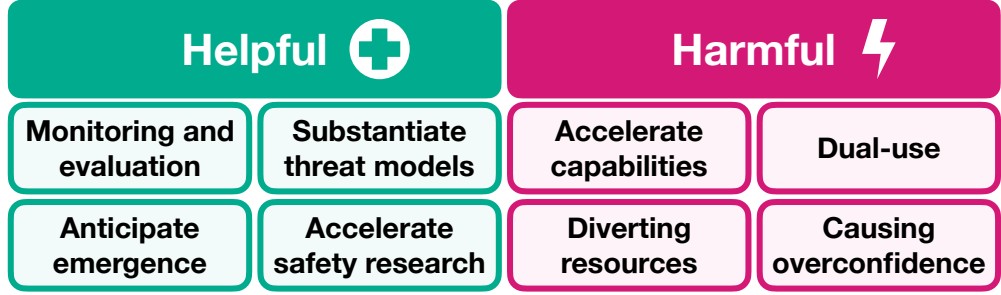

Figure 11: Potential benefits and risks of mechanistic interpretability for AI safety.

**How Could Interpretability Promote AI Safety?** Gaining mechanistic insights into the inner workings of AI systems seems crucial for navigating AI safety as we develop more powerful models (Nanda, 2022e). Interpretability tools can provide an understanding of artificial cognition, the way AI systems process information and make decisions, which offers several potential benefits:

Mechanistic interpretability could accelerate AI safety research by providing richer feedback loops and grounding for model evaluation (Casper, 2023). It may also help anticipate emergent capabilities, such as the emergence of new skills or behaviors in the model before they fully manifest (Wei et al., 2022; Steinhardt, 2023; Nanda et al., 2023a; Barak et al., 2022). This relates to studying the incremental development of internal structures and representations as the model learns (Section 5.2). Additionally, interpretability could substantiate theoretical risk models with concrete evidence, such as demonstrating *inner misalignment* (when a model's behavior deviates from its intended goals) or *mesa-optimization* (the emergence of unintended subagents within the model) (Hubinger et al., 2019; von Oswald et al., 2023). It may also trigger normative shifts within the AI community toward rigorous safety protocols by revealing potential risks or concerning behaviors (Hubinger, 2019a).

Regarding specific AI risks (Hendrycks et al., 2023), interpretability may prevent malicious misuse by locating and erasing sensitive information stored in the model (Meng et al., 2022a; Nguyen et al., 2022). It could reduce competitive pressures by substantiating potential threats, promoting organizational safety cultures, and supporting AI alignment (ensuring AI systems pursue intended goals) through better monitoring and evaluation (Hendrycks & Mazeika, 2022). Interpretability can provide safety filters for every stage of training: before training by deliberate design (Hubinger, 2019a), during training by detecting early signs of misalignment and potentially shifting the distribution towards alignment (Hubinger, 2022; Sharkey, 2022), and after training by rigorous evaluation of artificial cognition for honesty (Burns et al., 2023; Zou et al., 2023) and screening for deceptive behaviors (Park et al., 2023b).

The emergence of *internal world models* in LLMs, as posited by the *simulation* hypothesis, could have significant implications for AI alignment research. Finding an internal representation of human values and aiming the AI system's objective may be a trivial way to achieve alignment (Wentworth, 2022), especially if the world model is internally separated from notions of goals and agency (Ruthenis, 2022). In such cases, world model interpretability alone may be sufficient for alignment (Ruthenis, 2023).

Conditioning pre-trained models is considered a comparatively safe pathway towards general intelligence, as it avoids directly creating agents with inherent goals or agendas (Jozdien, 2022; Hubinger et al., 2023). However, prompting a model to simulate an actual agent, such as "You are a superintelligence in 2035 writing down an alignment solution," could inadvertently lead to the formation of internal agents (Hubinger et al., 2023). In contrast, reinforcement learning tends to create agents by default (Casper et al., 2023a; Ngo et al., 2022).

The *prediction orthogonality* hypothesis suggests that prediction-focused models like GPT can simulate agents with potentially misaligned objectives (janus, 2022). Although GPT may lack genuine agency or intentionality, it may produce outputs that simulate these qualities (Bereska & Gavves, 2023; Shanahan et al., 2023). This underscores the need for careful *oversight* and, better yet, using mechanistic interpretability to search for internal agents or their constituents, such as optimization or search processes – an endeavor known as *searching for search* (NicholasKees & janus, 2022; Jenner et al., 2024).

Mechanistic interpretability integrates well into various AI alignment agendas, such as understanding existing models, controlling them, making AI systems solve alignment problems, and developing alignment theories (technicalities & Stag, 2023; Hubinger, 2020). It could enhance strategies like detecting *deceptive alignment* (hypothetical when a model ensures to appear aligned as to pursue misaligned goals without raising suspicion) (Park et al., 2023b), *eliciting latent knowledge* from models (Christiano et al., 2021), and enabling better scalable *oversight*, such as in *iterative distillation and amplification* (Chan, 2023). A high degree of understanding may even allow for *well-founded AI* approaches (AI systems with provable guarantees) (Tegmark & Omohundro, 2023) or *microscope AI* (extract world knowledge from the model without letting the model take actions) (Hubinger, 2019a). Furthermore, comprehensive interpretability itself may be an alignment strategy if we can identify internal representations of human values and guide the model to pursue those values by *retargeting an internal search process* (Wentworth, 2022). Ultimately, *understanding and control are intertwined*, and deeper understanding can control AI systems more reliably.

However, there is a spectrum of potential misalignment risks, ranging from acute, *model-centric* issues to gradual, *systemic* concerns (Kulveit, 2024). While mechanistic interpretability may address risks stemming directly from model internals – such as deceptive alignment or sudden capability jumps – it may be less helpful

for tackling broader systemic risks like the emergence of misaligned economic structures or novel evolutionary dynamics (Hendrycks, 2023b). The multi-scale risk landscape calls for a balanced research portfolio to minimize risk, where research on governance, complex systems, and multi-agent simulations complements mechanistic insights and model evaluations. The perceived utility of mechanistic interpretability for AI safety largely depends on researchers' priors regarding the likelihood of these different risk scenarios.

**How Could Mechanistic Insight be Harmful?** Mechanistic interpretability research could accelerate AI capabilities, potentially leading to the development of powerful AI systems that are misaligned with human values, posing significant risks (Soares, 2023; Kross, 2023; Hendrycks & Mazeika, 2022). While historically, interpretability research had little impact on AI capabilities, recent exceptions like discoveries about scaling laws (Hoffmann et al., 2022), architectural improvements inspired by studying induction heads (Olsson et al., 2022; Fu et al., 2023a; Poli et al., 2023; Schuster et al., 2022), and efficiency gains inspired by the logit lens technique (Schuster et al., 2022) demonstrated its potential to enhance capabilities. Scaling interpretability research may necessitate automation (Conmy et al., 2023; Bills et al., 2023), potentially enabling rapid self-improvement of AI systems (RicG, 2023). Some researchers recommend selective publication and focusing on lower-risk areas to mitigate these risks (Hobbhahn & Chan, 2023; Shovelain & McKernon, 2023; Elhage et al., 2022b; Nanda et al., 2023a).

Mechanistic interpretability also poses dual-use risks, where the same techniques could be used for both beneficial and harmful purposes. Fine-grained editing capabilities enabled by interpretability could be used for *machine unlearning* (removing private data or dangerous knowledge from models) (Guo et al., 2024; Sun et al., 2024; Nguyen et al., 2022; Pochinkov & , 2023) but could be misused for censorship. Similarly, while interpretability may help improve adversarial robustness (Räuker et al., 2023), it may also facilitate the development of stronger adversarial attacks (Mu & Andreas, 2020; Casper et al., 2023b).

Misunderstanding or overestimating the capabilities of interpretability techniques can divert resources from critical safety areas or lead to overconfidence and misplaced trust in AI systems (Charbel-Raphaël, 2023; Casper, 2023). Robust evaluation and benchmarking (Section 8.2) are crucial to validate interpretability claims and reduce the risks of overinterpretation or misinterpretation.

# 7 Challenges

## 7.1 Research Issues

**Need for Comprehensive, Multi-Pronged Approaches.** Current interpretability research often focuses on individual techniques rather than combining complementary approaches. To achieve a holistic understanding of neural networks, we propose utilizing a diverse interpretability toolbox that integrates multiple methods (see also Section 4.4), such as: *(i.)* Coordinating observational (*e.g.*, probing, logit lens) and interventional methods (*e.g.*, activation patching) to establish causal relationships. *(ii.)* Combining feature-level analysis (*e.g.*, sparse autoencoders) with circuit-level interventions (*e.g.*, path patching) to uncover representation-mechanism interplay. *(iii.)* Integrating intrinsic interpretability approaches with post-hoc analysis for robust understanding.

For example, coordinated methods could be used for *reverse engineering* trojaned behaviors (Casper et al., 2023c), where observational techniques identify suspicious activations, interventional methods isolate the relevant circuits, and intrinsic approaches guide the design of more robust architectures.

**Cherry-Picking and Streetlight Interpretability.** Another concerning pattern is the tendency to cherry-pick results, relying on a small number of convincing examples or visualizations as the basis for an argument without comprehensive evaluation (Räuker et al., 2023). This amounts to publication bias, showcasing an unrealistic highlight reel of best-case performance. Relatedly, many interpretability techniques are primarily evaluated on small toy models and tasks (Chughtai et al., 2023; Elhage et al., 2022b; Jermyn et al., 2022; Chen et al., 2023b), risking missing critical phenomena that only emerge in more realistic and diverse contexts. This focus on cherry-picked results from toy models is a form of *streetlight interpretability* (Casper, 2023), examining AI systems under only ideal conditions of maximal interpretability.

## 7.2 Technical Limitations

**Scalability Challenges and Risks of Human Reliance.** A critical hurdle is demonstrating the scalability of mechanistic interpretability to real-world AI systems across model size, task complexity, behavioral coverage, and analysis efficiency (Elhage et al., 2022b; Scherlis et al., 2023). Achieving a truly comprehensive understanding of a model's capabilities in all contexts is daunting, and the time and compute required must scale tractably. Automating interpretability techniques is crucial, as manual analysis quickly becomes infeasible for large models. The high human involvement in current interpretability research raises concerns about the scalability and validity of human-generated model interpretations. Subjective, inconsistent human evaluations and lack of ground-truth benchmarks are known issues (Räuker et al., 2023). As models scale, it will become increasingly untenable to rely on humans to hypothesize about model mechanisms manually. More work is needed on automating the discovery of mechanistic explanations and translating model weights into human-readable computational graphs (Elhage et al., 2022b), but progress on that front may also come from outside the field (Lu et al., 2024).

**Obstacles to Bottom-Up Interpretability.** There are fundamental questions about the tractability of fully *reverse engineering* neural networks from the bottom up, especially as models become more complex (Hendrycks, 2023a). Models may learn internal representations and algorithms that do not cleanly map to human-understandable concepts, making them difficult to interpret even with complete transparency (McGrath et al., 2022). This gap between human and model ontologies may widen as architectures evolve, increasing opaqueness (Hendrycks et al., 2022). Conversely, model representations might naturally converge to more human-interpretable forms as capability increases (Hubinger, 2019a; Feng & Steinhardt, 2023).

**Analyzing Models Embedded in Environments.** Real-world AI systems embedded in rich, interactive environments exhibit two forms of in-context behavior that pose significant interpretability challenges beyond understanding models in isolation. Externally, models may dynamically adapt to and reshape their environments through in-context learning from the interactions and feedback loops with their external environment (Leahy, 2023). Internally, the *hydra effect* demonstrates in-context reorganization, where models flexibly reorganize their internal representations in a context-dependent manner to maintain capabilities even after ablating key components (McGrath et al., 2023). These two instances of in-context behavior – external adaptation to the environment and internal self-reorganization – undermine interpretability approaches that assume fixed *circuits*. For models deeply embedded in rich real-world settings, their dynamic coupling with the external world via in-context environmental learning and their internal in-context representational reorganization make strong interpretability guarantees difficult to attain through analysis of the initial model alone.

**Adversarial Pressure Against Interpretability.** As models become more capable through increased training and optimization, there is a risk they may learn deceptive behaviors that actively obscure or mislead the interpretability techniques meant to understand them. Models could develop adversarial "mind-reader" components that predict and counteract the specific analysis methods used to interpret their inner workings (Sharkey, 2022; Hubinger, 2022). Optimizing models through techniques like gradient descent could inadvertently make their internal representations less interpretable to external observers (Hubinger, 2019b; Fu et al., 2023b; von Oswald et al., 2023). In extreme cases, a highly advanced AI system singularly focused on preserving its core objectives may directly undermine the fundamental assumptions that enable interpretability methods in the first place.

These adversarial dynamics, where the capabilities of the AI model are pitted against efforts to interpret it, underscore the need for interpretability research to prioritize worst-case robustness rather than just average-case scenarios. Current techniques often fail even when models are not adversarially optimized. Achieving high confidence in fully understanding extremely capable AI models may require fundamental advances to make interpretability frameworks resilient against an intelligent system's active deceptive efforts.

# 8 Future Directions

Given the current limitations and challenges, several key research problems emerge as critical for advancing mechanistic interpretability. These problems span four main areas: emphasizing conceptual clarity (Section 8.1), establishing rigorous standards (Section 8.2), improving the scalability of interpretability techniques (Section 8.3), and expanding the research scope (Section 8.4). Each subsection presents specific research questions and challenges that need to be addressed to move the field forward.

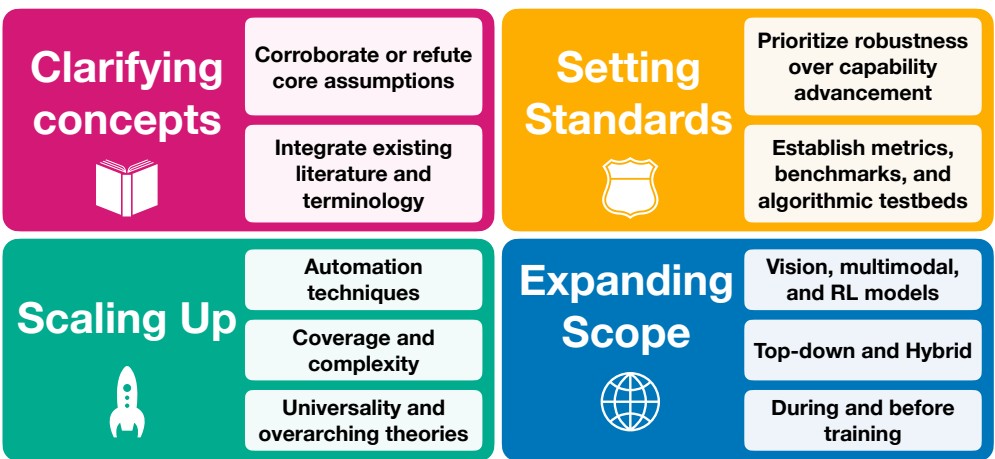

Figure 12: Roadmap for advancing mechanistic interpretability research, highlighting key strategic directions.

## 8.1 Clarifying Concepts

**Integrating with Existing Literature.** To mature, mechanistic interpretability should embrace existing work, using established terminology rather than reinventing the wheel. Diverging terminology inhibits collaboration across disciplines. Presently, the terminology used for mechanistic interpretability partially diverges from mainstream AI research (Casper, 2023). For example, while the mainstream speaks of *distributed representations* (Hinton, 1984; Olah, 2023) and the goal of *disentangled* representations (Higgins et al., 2018; Locatello et al., 2019), the mechanistic interpretability literature refers to the same phenomenon as *polysemanticity* (Scherlis et al., 2023; Lecomte et al., 2023; Marshall & Kirchner, 2024) and *superposition* (Elhage et al., 2022b; Henighan et al., 2023). Using common language invites "accidental" contributions and prevents isolating mechanistic interpretability from broader AI research.

Mechanistic interpretability relates to many other fields in AI research, including compressed sensing (Elhage et al., 2022b), modularity, adversarial robustness, continual learning, network compression (Räuker et al., 2023), neurosymbolic reasoning, trojan detection, and program synthesis (Casper, 2023; Michaud et al., 2024), and causal representation learning. These relationships can help develop new methods, metrics, benchmarks, and theoretical frameworks. For instance:

*i.)* **Neurosymbolic Reasoning and Program Synthesis**: Mechanistic interpretability aims for *reverse engineering* neural networks by converting their weights into human-readable algorithms. This endeavor can draw inspiration from neurosymbolic reasoning (Riegel et al., 2020) and program synthesis. Techniques like creating programs in domain-specific languages (Verma et al., 2019b;a; Trivedi et al., 2021), extracting decision trees (Zhang et al., 2019) or symbolic causal graphs (Ren et al., 2023) from neural networks align well with the goals of mechanistic interpretability. Adopting these approaches can extend the toolkit for reverse engineering AI systems.

*ii.)* **Causal Representation Learning**: Causal Representation Learning (CRL) aims to discover and disentangle underlying causal factors in data (Schölkopf et al., 2021), complementing mechanistic interpretability's goal of understanding causal structures within neural networks. While mechanistic interpretability typically examines individual *features* and *circuits*, CRL offers a framework for

understanding high-level causal structures. CRL techniques could enhance interpretability by identifying causal relationships between neurons or layers (Bengio et al., 2019; Ke et al., 2021), potentially revealing model reasoning. Its focus on interventions and counterfactuals (Pearl & Mackenzie, 2018; Peters et al., 2017) could inspire new methods for probing model internals (Goyal et al., 2020; Besserve et al., 2019). CRL's emphasis on learning invariant representations (Peters et al., 2015; von Kügelgen et al., 2019) could guide the search for robust features, while its approach to transfer learning (Rojas-Carulla et al., 2018; Magliacane et al., 2018) could inform studies into model generalization.

*iii.)* **Trojan Detection**: Detecting deceptive alignment models is a key motivation for inspecting model internals, as – by definition – deception is not salient from observing behavior alone (Casper et al., 2024). However, quantifying progress is challenging due to the lack of evidence for deception as an emergent capability in current models (Steinhardt, 2023), apart from *sycophancy* (Sharma et al., 2023; Denison et al., 2024) and theoretical evidence for *deceptive inflation* behavior (Lang et al., 2024). Detecting trojans (or backdoors) (Hubinger et al., 2024) implanted via data poisoning could be a proxy goal and proof-of-concept. These trojans simulate *outer misalignment* (where the model's behavior is misaligned with the specified reward function or objectives due to poorly defined or incorrect reward signals) rather than *inner misalignment* such as deceptive alignment (where the model appears aligned with the specified objectives but internally pursues different, misaligned goals). Moreover, activating a trojan typically results in an immediate change of behavior, while deception can be subtle, gradual, and, at first, entirely internal. Nevertheless, trojan detection can still provide a practical testbed for benchmarking interpretability methods (Maloyan et al., 2024).

*iv.)* **Adversarial Robustness**: There is a duality between interpretability and adversarial robustness (Elhage et al., 2022b; Räuker et al., 2023; Bereska, 2024). More interpretable models tend to be more robust against adversarial attacks (Jyoti et al., 2022), and vice versa, adversarially trained models are often more interpretable (Engstrom et al., 2019). For instance, techniques like input gradient regularization have been shown to simultaneously improve the interpretability of saliency maps and enhance adversarial robustness (Ross & Doshi-Velez, 2017; Du et al., 2021). Furthermore, interpretability tools can help create more sophisticated adversaries (Carter et al., 2019; Casper et al., 2021), improving our understanding of model internals. Viewing adversarial examples as inherent neural network *features* (Ilyas et al., 2019) rather than bugs also hints at alien features beyond human perception. Connecting mechanistic interpretability to adversarial robustness thus promises ways to gain theoretical insight, measure progress (Casper, 2023), design inherently more robust architectures (Fort & Lakshminarayanan, 2024), and create interpretability-guided approaches for identifying (and mitigating) adversarial vulnerabilities (García-Carrasco et al., 2024).

More details on the interplay between interpretability, robustness, modularity, continual learning, network compression, and the human visual system can be found in the review by Räuker et al. (2023).

**Corroborate or Refute Core Assumptions.** Features are the fundamental units defining neural representations and enabling mechanistic interpretability's bottom-up approach (Chan, 2023), but defining them involves assumptions requiring scrutiny, as they shape interpretations and research directions. Questioning hypotheses by seeking additional evidence or counter-examples is crucial.

The *linear representation* hypothesis treats activation directions as features (Park et al., 2023a; Nanda et al., 2023b; Elhage et al., 2022b), but the emergence and necessity of linearity is unclear – is it architectural bias or inherent? Stronger theory justifying linearity's necessity or counter-examples like autoencoders on uncorrelated data without intermediate linear layers (Elhage et al., 2022b) are needed. An alternative lens views features as polytopes from piecewise linear activations (Black et al., 2022), questioning if direction simplification suffices or added polytope complexity aids interpretability.

The *superposition* hypothesis suggests that *polysemantic* neurons arise from the network compressing and representing many features within its limited set of neurons (Elhage et al., 2022b), but polysemanticity can also occur incidentally due to redundancy (Lecomte et al., 2023; Marshall & Kirchner, 2024; McGrath et al., 2023). Understanding superposition's role could inform mitigating polysemanticity via regularization

(Lecomte et al., 2023). Superposition also raises open questions like operationalizing *computation in superposition* (Vaintrob et al., 2024; Hänni et al., 2024), *attention head superposition* (Elhage et al., 2022b; Jermyn et al., 2023; Lieberum et al., 2023; Gould et al., 2023), representing feature clusters (Elhage et al., 2022b), connections to adversarial robustness (Elhage et al., 2022b; García-Carrasco et al., 2024; Bloom & Bailey, 2023), anti-correlated feature organization (Elhage et al., 2022b), and architectural effects (Nanda, 2023a).

## 8.2 Setting Standards

**Prioritizing Robustness over Capability Advancement.** As the mechanistic interpretability community expands, it is essential to maintain the norm of not advancing AI capabilities while simultaneously establishing metrics necessary for the field's progress (Räuker et al., 2023). Researchers should prioritize developing comprehensive tools for analyzing the worst-case performance of AI systems, ensuring robustness and reliability in critical applications. This includes focusing on adversarial tasks, such as backdoor detection and removal (Lamparth & Reuel, 2023; Hubinger et al., 2024; Wu et al., 2022a), and evaluating the accuracy of explanations in producing adversarial examples (Goldowsky-Dill et al., 2023).

**Establishing Metrics, Benchmarks, and Algorithmic Testbeds.** A central challenge in mechanistic interpretability is the lack of rigorous evaluation methods. Relying solely on intuition can lead to conflating hypotheses with conclusions, resulting in cherry-picking and optimizing for best-case rather than average or worst-case performance (Rudin, 2019; Miller, 2019; Räuker et al., 2023; Casper, 2023). Current ad hoc practices and proxy measures (Doshi-Velez & Kim, 2017) risk over-optimization (Goodhart's law – *When a measure becomes a target, it ceases to be a good measure*). Distinguishing correlation from causation is crucial, as interpretability illusions demonstrate that visualizations may be meaningless without causal linking (Bolukbasi et al., 2021; Friedman et al., 2023a; Olah et al., 2017).

To advance the field, rigorous evaluation methods are needed. These should include: *(i)* assessing out-of-distribution inputs, as most current methods are only valid for specific examples or datasets (Räuker et al., 2023; Ilyas et al., 2019; Mu & Andreas, 2020; Casper et al., 2023c; Burns et al., 2023); *(ii)* controlling systems through edits, such as implanting or removing trojans (Mazeika et al., 2022) or targeted editing (Ghorbani & Zou, 2020; Dai et al., 2022; Meng et al., 2022a;b; Bau et al., 2018; Hase et al., 2023); *(iii)* replacing components with simpler reverse-engineered alternatives (Lindner et al., 2023); and *(iv)* comprehensive evaluation through replacing components with hypothesized circuits (Quirke et al., 2024).

Algorithmic testbeds are essential for evaluating faithfulness (Jacovi & Goldberg, 2020; Hanna et al., 2024) and falsifiability (Leavitt & Morcos, 2020). Tools like Tracr (Lindner et al., 2023) can provide ground truth labels for benchmarking search methods (Goldowsky-Dill et al., 2023), while toy models studying superposition in computation (Vaintrob et al., 2024) and transformers on algorithmic tasks can quantify sparsity and test intrinsic methods. Recently, Thurnherr & Scheurer (2024); Gupta et al. (2024) introduced datasets of transformer weights with known circuits for evaluating mechanistic interpretability techniques.

## 8.3 Scaling Techniques

**Broader and Deeper Coverage of Complex Models and Behaviors.** A primary goal in scaling mechanistic interpretability is pushing the Pareto frontier between model and task complexity and the coverage of interpretability techniques (Chan, 2023). While efforts have focused on larger models, it is equally crucial to scale to more complex tasks and provide comprehensive explanations essential for provable safety (Tegmark & Omohundro, 2023; Dalrymple et al., 2024; Gross et al., 2024) and enumerative safety (Cunningham et al., 2024; Elhage et al., 2022b) by ensuring models won't engage in dangerous behaviors like deception. Future work should aim for thorough *reverse engineering* (Quirke & Barez, 2023), integrating proven modules into larger networks (Nanda et al., 2023a), and capturing sequences encoded in hidden states beyond immediate predictions (Pal et al., 2023). Deepening analysis complexity is also key, validating the realism of toy models (Elhage et al., 2022b) and extending techniques like path patching (Goldowsky-Dill et al., 2023; Liu et al., 2023a) to larger language models. The field must move beyond small transformers

on algorithmic tasks (Nanda et al., 2023a) and limited scenarios (Friedman et al., 2023a) to tackle more complex, realistic cases.

**Towards Universality.**   As mechanistic interpretability matures, the field must transition from isolated empirical findings to developing overarching theories and universal reasoning primitives beyond specific circuits, aiming for a comprehensive understanding of AI capabilities. While collecting empirical data remains valuable (Nanda, 2023f), establishing motifs, empirical laws, and theories capturing universal model behavior aspects is crucial. This may involve finding more circuits/features (Nanda, 2022a;c), exploring circuits as a lens for memorization/generalization (Hanna et al., 2023), identifying primitive general reasoning skills (Feng & Steinhardt, 2023), generalizing specific findings to model-agnostic phenomena (Merullo et al., 2023), and investigating emergent model generality across neural network classes (Ivanitskiy et al., 2023). Identifying universal reasoning patterns and unifying theories is key to advancing interpretability.

**Automation.**   Implementing automated methods is crucial for scaling interpretability of real-world state-of-the-art models across size, task complexity, behavior coverage, and analysis time (Hobbhahn, 2022). Manual circuit identification is labor-intensive (Lieberum et al., 2023), so automated techniques like circuit discovery and sparse autoencoders can enhance the process (Foote et al., 2023; Nanda, 2023b). Future work should automatically create varying datasets for understanding circuit functionality (Conmy et al., 2023), develop automated hypothesis search (Goldowsky-Dill et al., 2023), and investigate attention head/MLP interplay (Monea et al., 2023). Scaling sparse autoencoders to extract high-quality features automatically for frontier models is critical (Bricken et al., 2023). Still, it requires caution regarding potential downsides like AI iteration outpacing training (RicG, 2023) and loss of human interpretability from tool complexity (Doshi-Velez & Kim, 2017).

## 8.4   Expanding Scope

**Interpretability Across Training.**   While mechanistic interpretability of final trained models is a prerequisite, the field should also advance interpretability before and during training by studying learning dynamics (Nanda, 2022b; Elhage et al., 2022b; Hubinger, 2022). This includes tracking neuron development (Liu et al., 2021), analyzing neuron set changes with scale (Michaud et al., 2023), and investigating emergent computations (Quirke & Barez, 2023). Studying phase transitions could yield safety insights for *reward hacking* risks (Olsson et al., 2022).

**Multi-Level Analysis.**   Complementing the predominant bottom-up methods (Hanna et al., 2023), mechanistic interpretability should explore top-down and hybrid approaches, a promising yet neglected avenue. The top-down analysis offers a tractable way to study large models and guide microscopic research with macroscopic observations (Variengien & Winsor, 2023). Its computational efficiency could enable extensive "comparative anatomy" of diverse models, revealing high-level motifs underlying abilities. These motifs could serve as analysis units for understanding internal modifications from techniques like instruction fine-tuning (Ouyang et al., 2022) and reinforcement learning from human feedback (Christiano et al., 2017; Bai et al., 2022).

**New Frontiers: Vision, Multimodal, and Reinforcement Learning Models.**   While some mechanistic interpretability has explored convolutional neural networks for vision (Cammarata et al., 2021; 2020), vision-language models (Palit et al., 2023; Salin et al., 2022; Hilton et al., 2020), and multimodal neurons (Goh et al., 2021), little work has focused on vision transformers (Palit et al., 2023; Aflalo et al., 2022; Vilas et al., 2023; Pan et al., 2024). Future efforts could identify mechanisms within vision-language models, mirroring progress in unimodal language models (Nanda et al., 2023a; Wang et al., 2023).

Reinforcement learning (RL) is also a crucial frontier given its role in advanced AI training via techniques like reinforcement learning from human feedback (RLHF) (Christiano et al., 2017; Bai et al., 2022), despite potentially posing significant safety risks (Bereska & Gavves, 2023; Casper et al., 2023a). Interpretability of RL should investigate reward/goal representations (Mini et al., 2023; Colognese & Jozdien, 2023; Colognese, 2023; Bloom & Colognese, 2023; Bloom & Bailey, 2023), study circuitry changes from alignment algorithms (Prakash et al., 2024; Jain et al., 2023; Lee et al., 2024; Jain et al., 2024), and explore emergent subgoals

or proxies (Hubinger et al., 2019; Ivanitskiy et al., 2023) such as internal reward models (Marks et al., 2023b). While current state-of-the-art AI systems as prediction-trained LLMs are considered relatively safe (Hubinger et al., 2023), progress on interpreting RL systems may prove critical for safeguarding the next paradigm (Aschenbrenner, 2024).

## Acknowledgements

I am grateful for the invaluable feedback and comments from Leon Lang, Tim Bakker, Jannik Brinkmann, Can Rager, Louis van Harten, Jacqueline Bereska, Benjamin Shaffrey, Thijmen Nijdam, Alice Rigg, Arthur Conmy, and Tom Lieberum. Their insights substantially improved this work.

## Glossary

**circuits** Sub-graphs within neural networks consisting of *features* and the weights connecting them. Circuits can be thought of as *computational primitives* that perform understandable operations to produce (ideally interpretable) features from prior (ideally interpretable) features. Examples include circuits for detecting curves at specific orientations (Cammarata et al., 2020; 2021), continuing repeated patterns in text (Olsson et al., 2022), and resolving anaphoric references (Wang et al., 2023). While circuits can involve clearly interpretable features, the definition allows for intermediate representations that are less easily interpretable.. 3, 9, 10, 20, 26, 27, 35

**concepts** An abstract idea or representation derived from observations of the world. Concepts refer to the *natural abstractions* that a cognitive system, like a neural network, aims to capture and represent through its learned *features*, which may or may not align perfectly with human-defined concepts.. 4, 6, 7, 32, 34

**deceptive alignment** When a misaligned model aims to appear aligned to gain more power to take control once sufficiently powerful.. 24

**deceptive inflation** Theoretical result on deceptive behavior: policies produce trajectories that look better than they actually are from the human's perspective with limited observations to get higher reward signals during training. This deceptive behavior arises in reinforcement learning from human feedback when the human provides feedback based only on partial observations of the trajectories, while the policy has full state information during training (Lang et al., 2024).. 28

**disentangled** In disentangled representations, individual dimensions or components correspond to distinct, *independent factors of variation in the data*, rather than representing a tangled mixture of these factors.. 4, 9, 15, 16, 20, 27, 32, 33

**eliciting latent knowledge** Developing strategies to make a machine learning model explicitly report latent facts or knowledge embedded in its parameters, especially in cases where the model's output is untrusted (Christiano et al., 2021). This involves finding patterns in neural network activations that track the true state of the world (Mallen & Belrose, 2023).. 24

**features** The fundamental units of how neural networks encode knowledge, which cannot be further decomposed into smaller, distinct *concepts*. Features are core components of a neural network's representation, analogous to how cells form the fundamental unit of biological organisms (Olah et al., 2020). The *superposition* hypothesis suggests an alternative definition: that features correspond to the *disentangled* concepts that a larger, sparser network with *sufficient capacity* would learn to represent with individual (*monosemantic*) neurons (Olah et al., 2020; Bricken et al., 2023).. 3, 4, 6–8, 10, 13, 16, 18, 20, 23, 27, 28, 32, 33, 35

**grokking** "Grokking refers to the surprising phenomenon of delayed generalization where neural networks, on certain learning problems, generalize long after overfitting their training set." (Liu et al., 2022a). 21, 22

**hydra effect** The phenomenon where models can internally self-repair and maintain capabilities even when key components are ablated, making it challenging to identify the relevant components underlying a particular behavior (McGrath et al., 2023).. 18, 26

**inner misalignment** Inner misalignment, or goal misgeneralization, occurs when an AI system develops goals or behaviors during training that are misaligned with the intended objectives despite a correctly specified reward signal.. 24, 28

**internal world models** Internal causal environment models formed within neural networks, implicitly emerging as a by-product of prediction (e.g., in large language models).. 11, 12, 24, 34

**irreducible** We adopt the notion of *features* as the fundamental units of neural network representations, such that features cannot be further decomposed into smaller, distinct factors. To make this more precise, we can formalize the definition of features as irreducible input patterns following Engels et al. (2024): A feature $f$ of sparsity $s$ is a function that maps a subset of the input space (with probability $1 - s > 0$) into a higher-dimensional representational space. We say the feature is active on this subset. A feature $f$ is reducible into features $a$ and $b$ if there exists a transformation that decomposes $f$ into $a$ and $b$, such that the transformed distribution $p(a, b)$ is either:

1. Separable: $p(a, b) = p(a)p(b)$
2. A mixture: $p(a, b) = wp_1(a, b) + (1 - w)p_2(a, b)$ where $p_1$ is lower-dimensional.

Features are defined as irreducible patterns that cannot be decomposed into separable or mixture distributions via such transformations. This formalizes the notion that features form the fundamental atomic units underlying neural representations. Features that can be *disentangled* into statistically independent components (separable) or simpler lower-dimensional factors (mixtures) are not considered the core representational primitives. The key properties are that 1) features map from the input space to higher-dimensional representational spaces, 2) features are sparse and only activated on subsets of the input, and crucially, 3) features are irreducible and cannot be expressed as transformations of other statistically independent components.. 4

**iterative distillation and amplification** A technique for training AI systems by repeatedly distilling knowledge from a larger model into a smaller one while amplifying the smaller model's capabilities through feedback and interaction with humans.. 24

**linear representation** Features are directions in activation space, i.e., linear combinations of neurons.. 3, 8, 14, 28

**machine unlearning** Techniques for removing private data or dangerous knowledge from models.. 25

**mesa-optimization** The emergence of unintended subagents within a model with their own objectives, potentially misaligned with the original training objective.. 24

**microscope AI** Systems that extract and utilize knowledge from a model without allowing the model to take autonomous actions. This involves reverse engineering a trained model to understand its learned knowledge about the world, aiming to leverage this understanding directly without deploying the model in an operational capacity.. 24

**modularity** The property of an AI system being composed of distinct, semi-independent components or submodules that can be separately understood, modified, and recombined, rather than a monolithic, opaque structure.. 20, 21

**monosemantic** A neuron corresponding to a single concept. The intuition is that analyzing what inputs activate a given neuron reveals its associated semantic meaning or concept. In contrast to *polysemantic*.. 5, 8, 14, 16, 20, 32, 34

**motifs** Repeating patterns that emerge across models and tasks, manifesting as circuits, features, or higher-level behaviors from component interactions. Examples include curve detectors, induction circuits, and branch specialization. Motifs reveal common structures and mechanisms underlying neural network intelligence.. 3, 10, 21, 35

**natural abstractions** High-level summaries or descriptions of a system or environment learned and used by many cognitive systems. According to the *natural abstraction hypothesis* (Chan et al., 2023), a set of "natural" abstractions exist that represent redundantly encoded information in the world and tend to be learned by intelligent systems produced through local selection pressures. These natural abstractions form a relatively small, discrete set of concepts like "tree," "velocity," etc., that allow compact descriptions of the world while discarding many irrelevant low-level details.. 4, 11, 32

**outer misalignment** Outer misalignment, or reward hacking, occurs when the specified reward function or utility function fails to capture the desired objectives correctly. This leads the AI to optimize for behaviors that achieve high reward scores but are misaligned with the intended outcomes.. 28, 34

**oversight** (Scalable) oversight refers to the challenge of providing reliable supervision—through labels, reward signals, or critiques—to AI models, ensuring effectiveness even as models *surpass* human-level performance.. 23, 24

**polysemantic** Neurons that are associated with multiple, unrelated *concepts*, contradicting the interpretation of neurons as representational primitives and making it challenging to understand the information processing of neural networks. This term is derived from linguistic concepts of *polysemy* (Falkum & Vicente, 2015), and in the context of neural networks first introduced by Arora et al. (2018), who suggested that word embeddings of polysemous words may be stored as a *superposition* of vectors representing distinct meanings. Olah et al. (2020) first used the term *polysemanticity*, elaborating on the concept of *polysemantic* neurons as a challenge for mechanistic interpretability.. 5, 16, 28, 33

**prediction orthogonality** A model whose objective is prediction can simulate agents who optimize toward any objectives with any degree of optimality (janus, 2022).. 3, 12, 24

**privileged basis** In certain neural network representations, the basis directions formed by the individual neurons are architecturally distinguished from arbitrary directions in the activation space. This privileged basis makes it meaningful to analyze the properties and roles of individual neurons, as the architecture encourages features to align with these basis directions. Hence, a privileged basis is *necessary* but *not sufficient* for the formation of *monosemantic* neurons. (Elhage et al., 2022b).. 5

**representation engineering** A top-down approach to transparency research that treats representations as the fundamental unit of analysis, aiming to understand and control representations of high-level cognitive phenomena in neural networks like large language models. Representation engineering has two main areas: 1) Reading representations to probe and interpret their contents, and 2) Controlling representations to manipulate high-level concepts like honesty or morality (Zou et al., 2023).. 3, 9, 22

**reverse engineering** The process of deconstructing a neural network's computations to fully understand and specify its operations. This involves breaking down the network's functionality into explicit, interpretable components, potentially as clear and detailed as pseudocode.. 1, 3, 20, 21, 25–27, 29

**reward hacking** See *outer misalignment*.. 30

**simulacra** The text outputs generated by a predictive model simulating the causal processes underlying text creation. These outputs simulate coherent and contextually relevant language, sometimes exhibiting agentic behaviors or goals despite the predictive model itself lacking genuine agency or intentionality. Simulacra can be either *agentic*, mimicking intentional and persuasive language use, or *non-agentic*, merely generating descriptive text without simulated goals or agency (janus, 2022; Bereska & Gavves, 2023).. 12, 14

**simulation** The simulation hypothesis says that when scaled up sufficiently, predictive models will learn to simulate the real-world causal processes that generated their training data (janus, 2022). When these models are optimized for predictive accuracy on broad data distributions like natural language, they are incentivized to discover the underlying rules, physics, and semantics that govern the data to model and predict future observations effectively. This allows the models to go beyond just memorizing or pattern-matching their training sets, instead learning to simulate hypothetical scenarios, reason about counterfactuals, and exhibit behaviors characteristic of general intelligence – all as a byproduct of the drive for efficient compression and accurate prediction. The simulation hypothesis suggests these models will develop rich *internal world models* capturing the causal dynamics of the training distribution.. 3, 11, 12, 14, 24

**streetlight interpretability** Examining AI systems under only ideal conditions of maximal interpretability, risking missing critical phenomena that only emerge in more realistic and diverse contexts.. 25

**superposition** The superposition hypothesis suggests that neural networks can leverage high-dimensional spaces to represent more *features* than the actual count of neurons by encoding features in almost orthogonal directions (Elhage et al., 2022b).. 3, 5, 6, 16, 21, 28, 32, 34

**sycophancy** The tendency of models to generate responses that align with user beliefs rather than providing truthful information. This behavior, encouraged by human feedback used in fine-tuning, is observed in state-of-the-art AI assistants across various tasks (Sharma et al., 2023). Sycophancy arises because human preference judgments often favor responses that match users' views, leading to a preference for convincingly written sycophantic responses over correct ones.. 28

**universality** The universality hypothesis proposes the emergence of common *circuits* across neural network models trained on similar tasks and data distributions. A *stronger* form posits that these common circuits represent a set of fundamental computational *motifs* that neural networks gravitate towards when learning. The *weaker* version suggests that for a given task, dataset, and model architecture, an optimal way to solve the problem may exist, which different models will tend to converge towards, resulting in analogous circuits. The universality hypothesis implies that rather than each model learning arbitrary, unstructured representations, there is an underlying universality to the circuits that emerge, shaped by the learning task and inductive biases.. 3, 9, 10, 21, 22

**well-founded AI** Developing AI systems with provable safety guarantees about their behavior and alignment with human values through rigorous mathematical modeling and verification. (Tegmark & Omohundro, 2023; Dalrymple et al., 2024).. 24

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
