# OpenReview forum: "Mechanistic Interpretability for AI Safety - A Review"
_TMLR — Accepted by TMLR_

### Review · Reviewer_ijrU · 2024-05-21

**Summary Of Contributions:**

Mechanistic Interpretability (MI) is a new avenue of research that targets reverse engineering the functioning of deep learning models. In this paper, the authors review works in this area by analyzing the fundamental concepts and hypotheses, collecting the core methods, presenting evaluations and tested scenarios, discussing applications to AI safety, and describing possible future avenues. MI so far stands as an independent branch of Explainable AI (XAI) for bottom-up explainability,
comprising the study of intrinsic interpretable methods,
 the analysis of neural networks' dynamics,
and assessing post-hoc evaluations of trained models (especially LLMs).
The authors collected a series of works from these three branches to create a common ground for future research in MI.

**Audience:**

Yes

**Broader Impact Concerns:**

There is no major negative ethical implications concerning this work.

**Claims And Evidence:**

Yes

**Requested Changes:**

Authors should address the main weaknesses of their work:

1) Better formalization of the notions and concepts presented in their manuscript. It is crucial to fix the relevant parts, especially if this review appears to be the official reference for MI.

2) Taxonomy listing how MI approaches are related to one another, w.r.t. to axes the authors highlighted. This could strengthen the presentation, providing a concise overview of different works in MI.

I hope the authors will also take into account other suggestions if not all, which may improve connections to other fields of research.

**Strengths And Weaknesses:**

# Strenghts

**Necessity of a common ground for Mechanistic interpretability research.** MI developed over the past years as a detached research area, with its own terminology and challenges, necessitating formal links to other better-known research branches in AI, especially with XAI. The reference to date introducing the notion of MI is [Olah 2020], and often cited as the primary contribution in the area, which is a blog post not followed by any peer-reviewed publication clarifying the terminology and the specific challenges of this research field. Following, other blog posts also explored aspects of MI, introducing novel ideas and terminologies, often conflating with existing literature in XAI. To this end, the effort of the authors to recollect several papers under the same umbrella and set the core concepts and challenges is valuable and can serve as a basis for future works in MI. Moreover, cross-fertilization with other AI research is promoted and could benefit both MI and XAI researchers.

**Sound presentation of core ideas and useful taxonomy.** The authors contribute to pointing to the core concepts (features, circuits, motifs), and typical assumptions and hypotheses (superposition, linearity, universality, simulation). Core methods are divided into the observational vs interventional nature and into what aspects of training they target: pre/during/post- training. This taxonomy is helpful and does comprise many, if not all, works in MI.

**Relevance to AI safety.** Authors point to applications of MI to safety/trustworthy AI, which could be integrated with current LLMs research. The discussion touches on both helpful and harmful aspects of MI applied to safe AI, making MI a potential new avenue of research also for trustworthy AI.

**Future challenges.** Eventually, authors detect future challenges and research directions that can help improve current methodologies in MI. This is valuable when pointing to common standards that future research should share and to indicating problems and future avenues to the researchers in the field.

# Weaknesses

### Clarity
Several parts require more scrutiny and better formalization. While previous papers and blog posts avoid being specific in terminology and formalization, the paper should avoid adopting a confusing conceptualization.

**Definitions.** In section 3, the authors introduce the notions of features, circuits, and motifs. In the presentation, however, (human-understandable) concepts are often mentioned but not explained making somehow vague how should the various notions be interpreted. The reference to concepts keeps appearing in different parts of the text and appears to be rooted in intuition. However, this poses a degree of ambiguity in definitions. In section 3.1 the authors write *'' We adopt the notion of features as the smallest units of how neural networks encode knowledge, such that features cannot be further decomposed into smaller, distinct concepts.''* which becomes vague and cannot be parsed from the text. In this respect, it is not clear (Definition: Feature, Alternative) what are the disentangled concepts. This should be made clear, especially in light of the presupposed distinction between concept-based explanations and MI methods (Sec 2 - Mechanistic). Mentioning *patterns* in (Def: Motifs) is also ambiguous and not clear from the text. If a pattern is either a feature or a circuit, it should be specified.

Regarding the hypotheses, universality is not also crystal clear what is the meaning that should be attributed to *analogous*: the same, or similar (up to what)? The definition of simulation is also a bit vague, regarding what it means to ''_simulate the underlying causal process of textual data_'' and what it means to ''_optimize strongly the model_''. As a stylistic note, it would be helpful to separate into a different subsection the notions of features and circuits and the discussion on the last three hypotheses (as mentioned at the beginning of section 3).
I am also not convinced that violations of the linearity hypothesis have not been observed yet: it depends on the context and clear counter-examples of some linear notion of features are present in the literature [1].  If some sort of linearity of the features holds, it should be specified in what domains and contexts, e.g. [2].


**Methods and Evaluations.** In the presentation of the core methods, at the attributional level methods based on (structured) probing have connections to XAI research. The review comprehends those that fall within MI, but it is not clear where the line should be drawn between XAI methods of this sort and MI methods. In light of the discussion in (Sec 4.1 - Structured Probes) regarding the ''_complementarity to MI efforts_''  it would be better to make clear the main differences between methods that use probes in MI and XAI. Feature Disentanglement is also mentioned only w.r.t. sparse autoencoders, but there is a rich literature in this respect, which should be mentioned in the potential future avenues of research, e.g. [1].
The presentation of Activation Patching can be improved. The explanation of the standard protocol can be made more precise by referring to Fig 7, and the example with A-OR-B and  A-AND-B cannot be understood entirely.

Also, it is not clear why methods based on causal abstractions and causal scrubbing are not discussed in Sec 4.2 and are related to rigorous hypothesis testing. This should be explained in the text.

**Overall structure and taxonomy**
While the overall structure is reasonable and offers a good distinction between methods, it would be helpful to resume what are the axes of distinctions between MI methods. The authors mention that methods can be distinguished based on their causal nature (observational/interventional) and the phase of learning they study (pre/during/post). Locality and globality are also mentioned for post-hoc methods and their comprehensiveness. These constitute other axes among which methods can be regrouped as customary in XAI research, see [3].

### Minors

**Connection to other research areas.** It is clear that MI has several intersections with other research on XAI. The main difference is the bottom-up approach of MI methods, although, it is not clear if all methods respect this general idea (e.g.  MI methods based on causal abstractions and high-level concepts, called top-down, ref. Sec 8.2 - Obstacles to bottom-up interpretability). Causality also seems to play a central role, due to disentangled representations and to interventions. Thus, the intersection with other works on XAI based on concepts [4] and Causal Representation Learning (CRL) [5]  can be fruitful for both research fields and MI. Recent works in CRL address the emergence of the Linear Hypothesis [6] and provably learning of (causal) concepts in foundation models [7]. Similarly, causality and CRL have been applied to defining causally interpretable representations [8].  Other works in this direction are also particularly relevant to Mechanistic Interpretability [9].

**Listing key concepts and working assumptions.** It can be helpful to provide an enlarged illustration of how MI differs from other methods and list the key concepts and hypotheses that are peculiar to MI, compared to other methods in XAI. This can help facilitate researchers approaching the field from related areas.

[1] Challenging Common Assumptions in the Unsupervised Learning of Disentangled Representations - ICML, Locatello et al. (2019) \
[2] Relative representations enable zero-shot latent space communication - ICLR, Moschella et al. (2023) \
[3] Explainable AI: A Review of Machine Learning Interpretability Methods - Entropy, Linardatos et al. (2021)  \
[4] Interpretability Beyond Feature Attribution: Quantitative Testing with Concept Activation Vectors - ICML, Kim et al. (2018) \
[5] Towards Causal Representation Learning - IEEE, Scholkopf et al. (2021) \
[6] On the Origins of Linear Representations in Large Language Models - arxiv, Jiang et al. (2024) \
[7] Learning Interpretable Concepts: Unifying Causal Representation Learning and Foundation Models - arxiv, Rajendran et al. (2024) \
[8] Interpretability is in the Mind of the Beholder: A Causal Framework for Human-interpretable Representation Learning - Entropy, Marconato et al. (2023) \
[9] Impossibility theorems for feature attribution, PNAS, Bilodeau et al. (2024)

---

> ### Author Response · Authors · 2024-06-17
> **Addressing definitions and hypotheses**
>
> Thank you for your thorough and insightful review of our paper on Mechanistic Interpretability (MI). We greatly appreciate your feedback and suggestions, which will help us improve our work's clarity and quality. We are pleased that you find our effort to establish a common ground for MI research valuable and recognize the necessity of this review.
>
> # Clarifying the definitions of features, circuits, and motifs
> ## Features and Concepts
> Our revised manuscript will explicitly distinguish between "features" as the model's learned representations and "concepts" as abstractions of the real world. Features should be understood as the fundamental units of the model's internal representations, which may or may not align with human-interpretable concepts. MI aims to uncover these learned features and understand how the model processes information, even if it differs from human intuition.
>
> We will provide more context on the natural abstraction hypothesis in the glossary (suggestion from reviewer Jfw9), which suggests that cognitive systems may converge on similar abstractions even if they are not easily interpretable to humans. This will help clarify the distinction between concept-based explanations, which focus on human-interpretable properties, and MI methods, which aim to uncover the model's internal representations.
>
> ## Motifs
> We have updated the definition of motifs to clarify that they refer to repeated circuit patterns rather than feature patterns across neural network models and tasks.
>
> We want to emphasize that our current definitions of features, circuits, and motifs should be considered working definitions that the field currently operates with to make progress. As MI advances, these definitions will likely require further refinement and formalization.
>
> # Clarifying the formulation of hypotheses
> ## Universality hypothesis
>
> We will clarify that the universality hypothesis posits a convergence in forming features and circuits across various models and tasks, with the degree of similarity depending on the level of abstraction. Different models might learn similar circuits for related tasks at a high level, even if the specific implementation details differ. Individual features might be less similar at a lower level due to differences in architecture or training data.
>
> We will distinguish between weak and strong versions of the universality hypothesis. The weak version suggests that models will converge on analogous solutions that adhere to common underlying principles, while the specific features and circuits may vary. The strong version proposes that the same core features and circuits will consistently arise across all models trained on similar tasks and data distributions.
>
> ##### Weak Universality
> > There are underlying principles governing how neural networks learn to solve certain tasks. Models generally converge on analogous solutions that adhere to the common underlying principles. However, the specific features and circuits implementing these principles can vary across models based on factors like hyperparameters, random seeds, and architectural choices.
>
> #### Strong Universality
> > The same core features and circuits will universally and consistently arise across all neural network models trained on similar tasks and data distributions and using similar techniques, reflecting a set of fundamental computational \term{motifs} that neural networks inherently gravitate towards when learning.
>
> Similarity can be measured by activating the same input examples for features and implementing the same mechanism for circuits, even if the specific wiring differs due to neural networks' natural equivariance.
>
> We propose the following new phrasing for the simulation hypothesis:
> #### Simulation and prediction orthogonality hypotheses
> > If a model for text prediction is optimized to sufficiently high predictive performance, it will form compressed internal representations that approximate and simulate the real-world causal generative processes underlying the text creation rather than surface statistics or correlations.
>
> We will acknowledge the ongoing debate about the role of inductive biases versus scale in learning high-level abstractions like causality from data alone.
>
> ## Linearity hypothesis
> We will discuss recent work (below) showing that linearity assumptions can be violated in certain contexts. We will also remove similarly quickly aging claims (as suggested by reviewer Jfw9).
>
> Counter-evidence:
> - Activation scale-dependent semantics of the decoded “features” in Templeton, A. et al. Scaling Monosemanticity. Transformer Circuits Thread (2024).
> - Circular features in Engels, J. et al. Not All Language Model Features Are Linear. CoRR (2024).
>
> Stylistic Note: As suggested by the reviewer, we will review the manuscript to ensure that the definitions and hypotheses are separated into different subsections for conceptual coherence.

---

> > ### Author Response · Authors · 2024-06-17
> > **Connections to related literature and taxonomy**
> >
> > ## Connections to XAI
> > We agree that MI has close ties to the broader field of XAI, particularly in probing techniques, and that there is much to be learned from the XAI literature. We view MI as an emerging subfield of XAI rather than a distinct field.
> >
> > In our revised manuscript, we will situate MI probing methods within the context of foundational XAI work on representation analysis and probing techniques, such as the influential works of Alain and Bengio (2016), Belinkov and Glass (2019), Tenney et al. (2019), Voita et al. (2019), and Dalvi et al. (2019), among others. We will clarify that while XAI probing has primarily focused on analyzing high-level concepts like linguistic representations, MI aims to push probing in a more mechanistic direction by uncovering the underlying computational processes and mechanisms within neural networks rather than just the predictive encoding of abstractions. This shift in goals towards uncovering mechanistic computations is a nuanced distinction rather than a clear-cut line between MI and XAI.
> >
> > ## Connection to disentanglement
> > We will expand our discussion of feature disentanglement to include relevant work from the broader literature on disentanglement beyond the works we already cited (Whittington et al., 2022; Higgins et al., 2018; O'Mahony et al., 2023; Locatelli et al., 2019). We will incorporate insights from additional key papers, such as:
> > Bengio et al. (2013) on the importance of disentangling factors of variation for representation learning
> > Mathieu et al. (2019) on disentangling disentanglement
> > Higgins et al. (2017) proposed the β-VAE framework for unsupervised learning of disentangled representations
> > Kim & Mnih (2018) on factorized latent representations in variational autoencoders
> >
> > We will explore other connections in more depth, discussing how recent advances in areas like CRL can inform MI research and how MI methods could be applied to problems in these other domains.
> >
> > ## Methods
> > Improving the presentation of activation patching and causal methods: We will revise the presentation of activation patching to make the standard protocol clearer and more accessible. We will use Figure 7 to illustrate transferring activations from a clean input to a corrupted input to isolate specific circuits. Additionally, we will provide a more intuitive example to illustrate the concepts of sufficiency and necessity in the context of Boolean logic circuits, making it easier for readers to grasp these key ideas.
> >
> > To highlight the role of causal methods in providing a more principled approach to evaluating explanations of neural network behavior, we will move the discussion of these methods to a separate subsection on hypothesis testing. This will emphasize the importance of causal interventions in establishing the relationships between internal components and model behavior, going beyond purely observational techniques.
> >
> > ## Taxonomy
> > We have found that current MI methods do not neatly group along these axes, particularly regarding the locality/globality and partial/comprehensive nature of their resulting interpretations. While MI research aims for global and comprehensive interpretability, most current methods and interpretations are narrow and local in scope. However, aggregating insights from multiple methods across multiple neurons or layers can contribute to a more global understanding of the model's behavior.
> >
> > We want to confirm with the reviewer if the following table aligns with their intended categorization of MI methods:
> >
> > | Method Category                                                       | Causal Nature   | Phase of Learning | Locality/Globality | Partial/Comprehensive  |
> > | --------------------------------------------------------------------- | --------------- | ----------------- | ------------------ | ---------------------- |
> > | Example-based, Feature-based methods                                  | Observational   | Post-hoc          | Local              | Partial                |
> > | (Structured) Probing, Logit Lens, Sparse autoencoders                 | Observational   | Post-hoc          | Local/Global       | Partial/Comprehensive  |
> > | Activation patching, (Direct) Path patching,                          | Interventional  | Post-hoc          | Local              | Partial                |
> > | Causal scrubbing, Causal abstraction, Locally consistent abstractions | Interventional  | Post-hoc          | Local/Global       | Comprehensive          |
> >
> > The categorization is based on the methods' general tendencies. Some methods, like probing and sparse autoencoders, can offer both local and global interpretability depending on the scope of the analysis.
> >
> > In conclusion, we thank the reviewer again for their detailed and thoughtful feedback. The suggestions for clarifying key concepts, improving the presentation of methods, and connecting MI to the broader literature will greatly enhance the clarity and impact of our work.

---

> > > ### Comment · Reviewer_ijrU · 2024-07-05
> > > **Response to authors**
> > >
> > > Thank you for your response. By reading your response to me and the other reviewers it seems that the following version of the paper addresses more or less the main points. I confirm the categorization of MI methods is sound.

---

### Review · Reviewer_Jfw9 · 2024-06-06

**Summary Of Contributions:**

This paper serves as a survey and introduction to the field of mechanistic interpretability. It outlines the terminology, provides definitions, overviews recent approaches based on these, and also provides a critique of aspects of the field (e.g., fundamental limitations of some methods, weaknesses in the rigour of some evaluations, etc).

**Audience:**

Yes

**Broader Impact Concerns:**

This paper is fairly well justified in its criticism of several aspects of the field and has a nuanced and balanced take on sensitive topics such as AI safety. I would urge the authors to take into account the above points though!

**Claims And Evidence:**

Yes

**Requested Changes:**

1. Simple change: numbering the definitions so it’s easy to refer to and go back to them.
2. More involved change: Having an index / glossary at the end that lists out all the terms defined could be helpful to readers that want to skim through these terms and link back to the context of where they were defined.
(e.g., format from several papers e.g., this paper: https://arxiv.org/abs/2311.04329)
3. Addressing all comments above!

**Strengths And Weaknesses:**

1. I think this paper is a well-constructed and researched survey paper that goes over definitions, hypothesis, methods and relevant aspects of the field of mechanistic interpretability. I have  some thoughts on making it a more comprehensive repository of information on mechanistic interpretability, and some things to change below:
2. It would be great to have a  more detailed anthology as in this paper. E.g., see this paper https://arxiv.org/abs/2311.04329 that links each definition to its place in the glossary allowing readers to go over and access all definitions easily.)
3. If each definition could be numbered and then linked in the glossary that would also help retrieve them more easily.
4. This paper interestingly falls in between two types of papers i.e., critique/position papers vs. a survey of all of the work in the field. Most of the paper is structured as a survey paper with definitions of important terms, hypotheses, algorithms and methods with the relevant citations needed for these. The latter part of the paper is more of a critique of aspects of the field. This division is great! However, in between some of the definitions/methods are critiques of certain methods that are not citable, or at least cited in this paper (they seem more subjective). For the sake of a comprehensive/unbiased survey paper, I would urge the authors to not provide subjective viewpoints in the descriptive/survey part. I’ve listed them all below.
5. An overarching concern with this type of survey paper is that the field is emerging so fast that a lot of the things in this paper are already outdated (e.g., claims such as “so far no one has done so and so in a certain task”) are not largely true anymore even at the time of reviewing this paper. It’s clearly necessary to outline what has and hasn’t been done until this point in time but I would urge the authors to reduce such claims.
6. For e.g., on page 8, “only relatively narrow behaviours like Python docstring formatting” have been explored so far”.
7. On page 8:” Hypothesis: Universality” it might be nice to also have the hypotheses linked in a similar glossary?
8. On page 9: on internal world models, there are several works on behavioural probing and mech interp that look into this that are not cited: https://openreview.net/forum?id=jE8xbmvFin, https://openreview.net/forum?id=gJcEM8sxHK, https://arxiv.org/abs/2106.00737
9. On page 12, re: “unsupervised methods can identify numerous features without a straightforward verification process”, can we add citations for this?
10. On page 14, the overview of the different activation patching approaches is great, but if this could be elaborated in some detail and cited that would be helpful!
11. Figure 7 is really great!
12. The terms, inner misalignment and mesa optimisation could be helpful to define in the anthology.
13, On page 22: “utilising a diverse interpretabiltiy toolbox”, how would the authors propose to do that? This sentence, without any explanation of what this implies doesn’t actually provide any insight or details into what could be done better.
14. On page 24: this might be hard to do but it would be extremely helpful to have a table that maps the definitions to each other re: the point about how terms are redefined and new terms/definitions really mean the same underlying thing or concept.
15. In section 2, RSA-like approaches could also be mentioned and cited here as a class of probing/interpretability techniques.
16. Re: monosemantic and polysmenatic neurons: can we cite where these terms originate from? (e.g., used in the linguistics literature here, Vicente and Falkum (2017), but it would be nice to have the original paper that used this both in semantics, and also in the interpretability field).
^maybe this paragraph should be phrased not in the context of neurons, but just talking about mono/polysemy, why it is important to make this distinction, and that it is about trying to find units that correspond to it. The next paragraph can then tie things together (as it does already).
17. The definition of features could be made more concrete—is just saying “fundamental unit” sufficient? Is the implication that features are monosemantic (whether human interpretable or otherwise?) Also it is currently unclear to me if they are implying that features equate to neurons?
18. When talking about world modeling and relating to AI safety: “emergent world models have significant implications … internal representation of human values” → it might be worth drawing the comparison between intrinsic values/human morals in the real world and the game/world representations that are typically studied e.g., in chess domains. These are usually smaller and less complex than a world that allows the incorporation of complex, nuances, human values and cultural information.
19. Activation patching is defined but not path patching and more complex variants. It might also be worth diving into how the patching is done (and the links to causal interventions).
20. “Causality as a theoretical foundation”: it would be great if this could/should be introduced a lot earlier? Especially when introducing patching techniques it is worth drawing the connection to causality and the theory behind a lot of this field of mechanistic interpretability methods. Even having some of the definitions of causal interventions from Pearl etc., would be helpful to readers new to this space!

---

> ### Author Response · Authors · 2024-06-21
> **Comprehensive Revisions Enhancing Structure, Content, and Accessibility**
>
> We sincerely appreciate your thorough and constructive feedback on our survey paper. Your comments have been instrumental in helping us improve the manuscript significantly. We have addressed your concerns and made extensive revisions to enhance the paper's clarity, structure, and content. Below, we detail our responses to each of your points:
>
> 1-3. We have implemented a detailed glossary system as suggested. Each definition is now numbered and linked in a comprehensive glossary, allowing readers to easily access and cross-reference terms throughout the paper.
>
> 4. We have restructured the paper to more clearly delineate between the survey and critique sections. Subjective viewpoints have been moved to more appropriate sections, such as relocating the discussion of simulation and prediction orthogonality hypotheses to the relevance section. We have also moved the subjective part on qualitative and quantitative evaluation to the future directions section.
>
> 5-6. We acknowledge the rapid pace of developments in the field. We have updated the manuscript with the latest findings, particularly regarding the linear representation hypothesis. We've also removed or revised "to date" claims to ensure the paper's longevity. We plan to continually update the arXiv and potentially create a website version to keep the paper as current as possible.
>
> 7. We have extended the glossary to include hypotheses, using the same format as for definitions. We plan to further extend this to include methods in the glossary before the camera-ready version.
>
> 8. We've incorporated the suggested citations on internal world models.
>
> 9. We have refactored the paragraph on unsupervised methods, basing our claims directly on the work by Farquhar et al. (2023) to provide a more robust foundation for our statements.
>
> 10. The activation patching section has been extensively revised, providing more detail, structure, and citations to offer a comprehensive overview of these approaches.
>
> 11. We appreciate your positive feedback on Figure 7. Nevertheless, we've opted to update the caption to make it more informative and better connected to the text.
>
> 12. We have added definitions for inner misalignment and mesa-optimization and other AI safety jargon to the glossary.
> 	12. b) We've revised the "Need for Comprehensive, Multi-Pronged Approaches" paragraph to clearly explain what we mean by utilizing a diverse interpretability toolbox.
> 13. We've created an overview figure at the beginning of the concepts section that illustrates the relationships between key concepts and hypotheses, addressing the need for a visual representation of how these terms interconnect.
>
> 14. We've added a sentence and citations on RSA-like approaches in Section 2, expanding our coverage of probing and interpretability techniques.
>
> 15. We've added information on the origins of the terms monosemantic and polysemantic to the glossary, providing context from both linguistics and interpretability literature.
>
> 16. We've fundamentally rewritten the section introducing features, adopting a more concrete definition based on recent work by Engels et al. (2024). This new definition clarifies the concept of features as irreducible units and their distinction from neurons.
>
> 17. We've added a paragraph discussing the limitations of current world modeling interpretability work, highlighting the gap between simple game domains and the complexity of real-world human values and cultural information.
>
> 18-19. We've introduced path patching and subspace activation patching, and repositioned the discussion of causality as a theoretical foundation earlier in the paper, providing a stronger context for the subsequent methods. We've also condensed the evaluation section and integrated its key points into other relevant subsections.
>
> These revisions have significantly improved the paper's structure, clarity, and comprehensiveness. We've addressed the need for better organization of concepts, more rigorous definitions, and up-to-date content. The addition of a comprehensive glossary, visual aids, and more explicit connections between concepts and methods enhances the paper's accessibility and usefulness as a reference in the rapidly evolving field of mechanistic interpretability.
>
> We thank you again for your insightful comments and hope that these extensive revisions meet your expectations. We believe these changes have substantially improved the quality and utility of our survey paper. Thank you again, for taking the time for such an in-depth review.

---

> > ### Comment · Reviewer_Jfw9 · 2024-07-08
> >
> > Thank you for the changes---I think the glossary helps make this a more accessible and easy-to-read resource as a survey paper. The addition of the other terms to be defined and the new figures re: other reviewers comments also help tie this together very nicely and I think the new and revised version is a very comprehensive and sound survey paper on mechanistic interpretability.

---

### Review · Reviewer_QHfE · 2024-06-08

**Summary Of Contributions:**

This paper focuses on investigating current methodologies in the field of mechanistic interpretability, which aims to uncover causal relationships and precise computations transforming inputs into outputs in a neural network. Towards this field, this paper presents its core concepts and hypotheses, explains methods and techniques, discusses evaluation, explores its relevance to AI safety, and points out challenges and future directions.

**Audience:**

Yes

**Broader Impact Concerns:**

Not applicable.

**Claims And Evidence:**

No

**Requested Changes:**

Please refer to the weakness. The four points are important to my final decision.

**Strengths And Weaknesses:**

**Strength**:

1. The topic of survey, mechanistic interpretability, is important and may attract interests from many researchers.

2. This paper does a extensive investigation for mechanistic interpretability. As far as I know, there are few survey papers on this topic.



**Weakness**:

My main concerns come from the organization of the survey paper. I understand that mechanistic interpretability is more complex than attribution/concept interpretability. But at current vision, I find it difficult to provide a coherent overview of this field. To improve the readability, I have following comments:

1. **There is short of overview figures/tables**, so as to (such as) clarify the connection between core concepts, the relationship between concepts and hypotheses, and so on. For example, how can we distinguish from features, semantics, neurons, and basis? What’s the connection between features, circuits, and motif? How does the Universality Hypothesis relate to these core concepts? I notice that the authors may give answers to some of the above questions in the texts, but this is not intuitive and impressive. It is crucial to help the readers form a clear overall picture.

2. This survey only lists core concepts and hypothesis, but **lacks a summary and refinement of key research problems** in the field of mechanism interpretability. Key research problems are essential to investigating a research field. Based on my understanding of this survey, key research problems may be “How a neural network  encode semantics? Is it encoded in neurons?”, “Whether a single neuron corresponds to an explicit semantic?”, and so on. I suggest the authors to have a deep thinking on summarizing research problems.

3. **There is a significant gap between the core concepts/research problems (Section 3) and methods/techniques (Section 4)**. The authors should give more details or examples on how to adopt these techniques in interpreting the mechanisms of neural networks.

4. I find the survey a little long and difficult to read. I suggest to condense the paper and polish the language, to stress the key point and make the statement more precise and understandable.

---

> ### Author Response · Authors · 2024-06-21
> **Revisions addressing readability, visual aids, clarifying research problems, and connections between concepts and methods**
>
> Thank you for your constructive feedback on our survey paper on mechanistic interpretability. We appreciate the time and effort you've invested in reviewing our work, and we have taken your comments seriously to improve the paper. We'd like to address each of your main concerns and explain how we've revised the manuscript in response.
>
> 1. **Lack of overview figures/tables:** We agree that visual aids can greatly enhance the readability and comprehension of complex topics. In response, we've added several new figures to the paper:
>
> - A comprehensive overview figure illustrating the key concepts and hypotheses in mechanistic interpretability, showing their relationships and organization.
> - A figure contrasting privileged and non-privileged bases, which helps clarify the distinctions between features, semantics, neurons, and basis.
> - A figure comparing observed models with hypothetical disentangled models, demonstrating the connections between features, circuits, and motifs.
>
> These additions should provide readers with a clearer overall picture of the field and its core concepts.
>
> 2. **Lack of summary and refinement of key research problems:** We appreciate your suggestion to make the key research problems more explicit. Upon review, we realized that our future directions section already contains numerous research problems, albeit implicitly framed. To address your concern, we have revised the introductory paragraph of the future directions section to more clearly highlight that each subsection presents key research problems in mechanistic interpretability. We have also added topic sentences to each paragraph that frames the content as specific research questions or problems. We believe this adjustment makes the research problems more apparent without altering the substantial content and structure of the section.
> 3. **Gap between core concepts/research problems and methods/techniques:**
> 	- To bridge this gap, we've rewritten the section on integrating observation and intervention methods. The revised version now includes explicit examples of how different techniques can be combined to investigate neural network mechanisms. For instance, we explain how sparse autoencoders can be used in conjunction with activation patching, and how the logit lens can be complemented with targeted interventions. We've also added recent examples of research that integrates multiple methods to provide a more comprehensive understanding of neural network behavior.
> 	- In addition, we've enhanced the connections between concepts and methods through the introduction of a glossary (as suggested by reviewer 2). This glossary serves to highlight key terms and their relationships throughout the paper. For example, in the pink box on sparse autoencoders, we've revised the content to emphasize how this method relates to core concepts such as polysemantic neurons, superposition, and disentangled features. The use of consistent terminology and visual cues (like the orange italic font for glossary items) makes it easier for readers to identify and track important concepts and their interconnections across different sections of the paper. Throughout the methods section we've highlighted how the techniques relate to the key concepts in this way.
> 4. **Length and readability**: We've made a concerted effort to condense and restructure the paper to improve its readability:
> 	- The core concepts section has been streamlined, with abridged discussions of the simulation and prediction hypotheses.
> 	- The activation patching section has been condensed.
> 	- The evaluation section has been significantly shortened and its key points integrated into other relevant subsections.
> 	- We've made numerous smaller edits throughout the paper to improve language clarity and presentation and will continue to improve this aspect for the camera-ready version.
>
> We believe these changes have made the paper more focused and easier to read while maintaining its comprehensive coverage of the field. We uploaded the revised document as pdf for your inspection.
>
> We hope that these revisions address your concerns and improve the overall quality and accessibility of our survey. We thank you again for your valuable feedback, which has helped us refine and enhance our work. We look forward to your thoughts on these changes.

---

### Decision · Action_Editor_znNE · 2024-07-24

**Recommendation:** Accept as is

**Comment:**

The paper structure and content has been appreciated by all reviewers. The aim of structuring the literature around interpretability has been found as a worthy goal, and reviewers considered that an omni-comprehensive survey could be impossible. At the same time, the initial revision of the manuscript has been criticized for missing some literature connections and references and for presenting dimensions that appeared to be detached one from another in the presentation. Lastly, some statements were imprecised and needed to be rewritten.

Authors took all comments into account and provided a revised paper that substantially improved presentation and readability, as well as better covering the interpretability landscape. All reviewers are willing to accept the paper and I agree with them. Furthermore, two reviewers agreed on proposing the Survey certification, as the revised version now looks pretty polished.

**Audience:**

Interpretability is a big hot topic in machine learning, and having a good starting point to disentangle the literature jungle around it is of certain interest for the TMLR audience.

**Claims And Evidence:**

This paper is a nice survey on modern approaches to interpret the behavior of machine learning models with more emphasis on approaches that elicit the mechanisms that supposedly lead these models to make a decision. The authors break down these approaches and illustrate the core concepts behind them (features, concepts, abstractions) and the paradigms they use (behavioral. mechanistic, etc).
On top of that, authors also provide a critique of several aspects in the literature of machine interpretability, such as fundamental limitations of certain approaches and potential lack of rigor when evaluating them.
In this format, the paper does not present any (controversial) claims that need further evidence to be supported.